

# Revision of the WMO/GAW CO₂ Calibration Scale

Bradley D. Hall[1], Andrew M. Crotwell[1,2], Duane R. Kitzis[1,2], Thomas Mefford[1,2], Benjamin R. Miller[3], Michael F. Schibig[4,5], and Pieter P. Tans[1]

[1] National Oceanic and Atmospheric Administration, Global Monitoring Laboratory, Boulder, CO, USA
[2] Cooperative Institute for Research in Environmental Sciences, University of Colorado, Boulder, CO, USA
[3] Previously with the Cooperative Institute for Research in Environmental Sciences, University of Colorado, Boulder, CO, USA
[4] Climate and Environmental Physics, University of Bern, Bern, Switzerland
[5] Oeschger Centre for Climate Change Research, University of Bern, Bern, Switzerland

*Correspondence to*: Brad Hall (Bradley.Hall@noaa.gov)

**Abstract**  The NOAA Global Monitoring Laboratory serves as the WMO/GAW Central Calibration Laboratory (CCL) for CO₂ and is responsible for maintaining the WMO/GAW mole fraction scale used as a reference within the WMO/GAW program. The current WMO-CO₂-X2007 scale is embodied by 15 aluminum cylinders containing natural air, with CO₂ mole fractions determined using the NOAA manometer from 1995 to 2006. We have made two minor corrections to historical manometric records: fixing an error in the applied second virial coefficient of CO₂, and accounting for loss of a small amount of CO₂ to materials in the manometer during the measurement process. By incorporating these corrections, extending the measurement records of the original 15 primary standards through 2015, and adding four new primary standards to the suite, we define a new scale, identified as WMO-CO₂-X2019. The new scale is 0.18 µmol mol⁻¹ (ppm) greater than the previous scale at 400 ppm CO₂. While this difference is small in relative terms (0.045%), it is significant in terms of atmospheric monitoring. All measurements of tertiary-level standards will be reprocessed to WMO-CO₂-X2019. The new scale is more internally consistent than WMO-CO₂-X2007 owing to revisions in propagation, and should result in an overall improvement in atmospheric data records traceable to the CCL.

## 1 Introduction

Measurements of the atmospheric distribution of carbon dioxide (CO₂) are essential to understanding sources and sinks of this powerful greenhouse gas. We need well calibrated measurements to track the history of the global abundance of CO₂ because it is the main driving force of man-made climate change. Small differences in the relative abundances of CO₂ and other trace gases observed at different locations, combined with information on atmospheric transport and mechanisms for land-atmosphere-ocean exchange can provide constraints on estimates of the sources and sinks of CO₂. Measurements are made at numerous sites around the globe in conjunction with the WMO Global Atmosphere Watch program and through regionally-coordinated programs (e.g. Integrated Carbon Observing System, ICOS).





Because the atmospheric gradients of $CO_2$ are small in the background atmosphere far from sources of pollution, the WMO/GAW has adopted a single reference scale, maintained and disseminated by a designated Central Calibration Laboratory

(CCL), on which to base all measurements made within the program. The quantity to be measured is the mole fraction of $CO_2$ in dry air ($\mu mol\ mol^{-1}$, abbreviated as ppm, from parts per million), because it is conserved when air expands or contracts or when water vapour is added or removed. The WMO community has set network compatibility goals for the measurements, 0.1 ppm in the northern and 0.05 ppm in the southern hemisphere, aimed at minimizing bias between measurement sites in the network (WMO, 2020). To help meet these stringent goals, the WMO/GAW community voted in their 1995 meeting for the

NOAA Climate Monitoring and Diagnostics Laboratory (subsequently known as the Global Monitoring Division, and currently as the Global Monitoring Laboratory) to serve as the Central Calibration Laboratory (CCL) for $CO_2$. The Scripps Institution of Oceanography (SIO) initially served in this capacity (Keeling et al., 1986) before responsibilities were transferred to NOAA. The WMO/GAW $CO_2$ calibration scale also serves as a reference linking other measurement programs, such as those involving aircraft and total column measurements to the surface measurement networks (Wunch et al., 2010;

Messerschmidt et al., 2011).

As the CCL for $CO_2$, NOAA maintains a set of 15 aluminum high-pressure gas cylinders containing modified natural air, with $CO_2$ spanning the range 250-520 ppm. $CO_2$ mole fractions were determined using an absolute method based on manometry (Zhao et al., 1997). These cylinders serve as primary standards and along with their assigned mole fractions constitute the

WMO-CO$_2$-X2007 mole fraction scale, where X is used to denote mole fraction, and 2007 is the year in which the assigned values were adopted (hereafter simplified to X2007). The scale is distributed in high-pressure aluminum cylinders containing natural air (tertiary-level standards) with value assignment made by comparison against secondary standards (also natural air), which are traceable to the primary standards. The CCL at NOAA is a designated institute of WMO, which is a signatory to the *Comité International des Poids et Mesures* Mutual Recognition Arrangement (CIPM-MRA). Accordingly, calibration and

measurement capabilities are listed in the Key Comparison Database maintained by the *Bureau International des Poids et Mesures* (BIPM) (http://kcdb.bipm.org/). It is through primary methods, such as manometry, and comparison to other validated methods, such as gravimetry, that traceability to the International System of Units (SI) is established (Milton, 2013).

Since 1995, primary standards have been measured every 2-3 years to develop a measurement history and monitor for possible

drift. Each measurement period is called an "episode". The X2007 scale was developed following the 2006 measurement episode (Tans et al., 2011). We have performed three measurement episodes since 2006 (2009, 2012, and 2015) to assess the X2007 assigned values using methods similar to those in use in 2006 (Zhao and Tans, 2006). Results from the 2009, 2012, and 2015 episodes were sufficiently close to the X2007 assignments that no updates to the scale have been made since 2007. While the X2007 scale has served the community well for more than a decade, there are some compelling reasons to update

the scale: 1) we discovered an error in the computer code used to reduce the manometer data; 2) we have improved our experimental methods in recent years, leading to a more accurate measure of $CO_2$ in the primary standards; 3) we would like



to expand the range of the WMO/GAW scale to 800 ppm to better constrain instrument response and also provide support for measurements obtained closer to emission sources, such as urban areas; and 4) we have recently developed a new measurement system used to transfer the scale to reference materials (Tans et al., 2017), which now allows us to harmonize the primary

standards and define the scale with higher precision than what can be done with a single standard (see section 6.0).

Here we introduce a revision of the WMO/GAW $CO_2$ scale, with the new scale identified as WMO-$CO_2$-X2019 (hereafter referred to as X2019), and describe its implementation. This article is organized as follows. We first provide some background on the manometric method. We then describe two corrections to previous manometric results. These include corrections to

rectify a calculation error related to the second viral coefficient of $CO_2$, and a correction for $CO_2$ absorption/adsorption to manometer surfaces (most likely O-rings) that occurs during the measurement process. The magnitude of the overall correction is small (~0.18 ppm at 400 ppm), but significant in terms of network compatibility goals (WMO, 2020). We have applied these corrections to 23 years of manometric measurements. By reassigning $CO_2$ mole fractions to previous and newly-introduced primary standards, we define the X2019 scale and explore differences between X2019 and X2007. We provide an estimate of

the uncertainty associated with $CO_2$ reference materials, updating the work of Zhao and Tans (2006). Finally, we propagate the X2019 scale to all reference materials analysed by the CCL and discuss the implementation of the X2019 scale.

## 2  The NOAA manometer

The manometric procedure is described in Zhao et al. (1997), and Zhao and Tans (2006). Briefly, the manometer consists of two glass volumes housed in a temperature-controlled oven, two glass traps for cryogenically extracting $CO_2$ from air and

purifying the $CO_2$, and devices to measure pressure and temperature (Fig. 1). During a measurement experiment, gas from a cylinder is loaded into the larger of the two volumes (large volume, ~6 L). After flushing the large volume for 10 min. at 200 mL min$^{-1}$ and allowing the gas temperature to equilibrate to oven temperature, the large volume is sealed off and the large volume temperature and pressure are recorded. The air sample is then pumped across the glass traps, which are held at liquid nitrogen temperature, to cryogenically extract the $CO_2$ from the air sample. The $CO_2$ is then purified (to remove $H_2O$) by

alternately freezing at L-$N_2$ temperature (~ -197 °C at 84 kPa) and warming to ~-67 °C. Finally, the purified $CO_2$ is cryogenically trapped into the smaller of the two volumes (~7 mL) and allowed to sublimate. The pressure and temperature of $CO_2$ in the small volume are recorded at ~30 s intervals as the $CO_2$ warms and equilibrates to the oven temperature.

The mole fraction of $CO_2$ is determined from measurements of pressure, temperature, and the ratio of the two volumes. The

volume ratio is determined by a gas expansion method using two additional volumes, also housed in the oven. A gas, usually air or nitrogen, is expanded into successive volumes, with P and T measured at each stage, to bridge the difference between small and large volumes (Zhao et al., 1997). The mole fraction of $CO_2$, $X_{CO2}$, is calculated using:





$$X_{CO2} = (\phi^{-1}) \frac{P_{CO2} T_{air}}{P_{air} T_{CO2}} (1 + A_1 - A_2) - X_{N2O} \qquad (1)$$

$$A_1 = \frac{P_{air} \beta_{air}}{R T_{air}}$$

$$A_2 = \frac{P_{CO2} \beta_{CO2}}{R T_{CO2}}$$

where T and P are the temperatures and pressures of air in the large volume (air) and nearly pure $CO_2$ in the small volume ($CO_2$), $\beta_{air}$ and $\beta_{CO2}$ are second virial coefficients, $R$ is the gas constant, $\Phi$ is the volume ratio (large/small), and $X_{N2O}$ is the mole fraction of $N_2O$ in the air sample (measured separately by gas chromatography with electron capture detection) (Hall *et al*., 2011). Equation (1) is an alternate form of eq. 8 from Zhao et al. (1997).

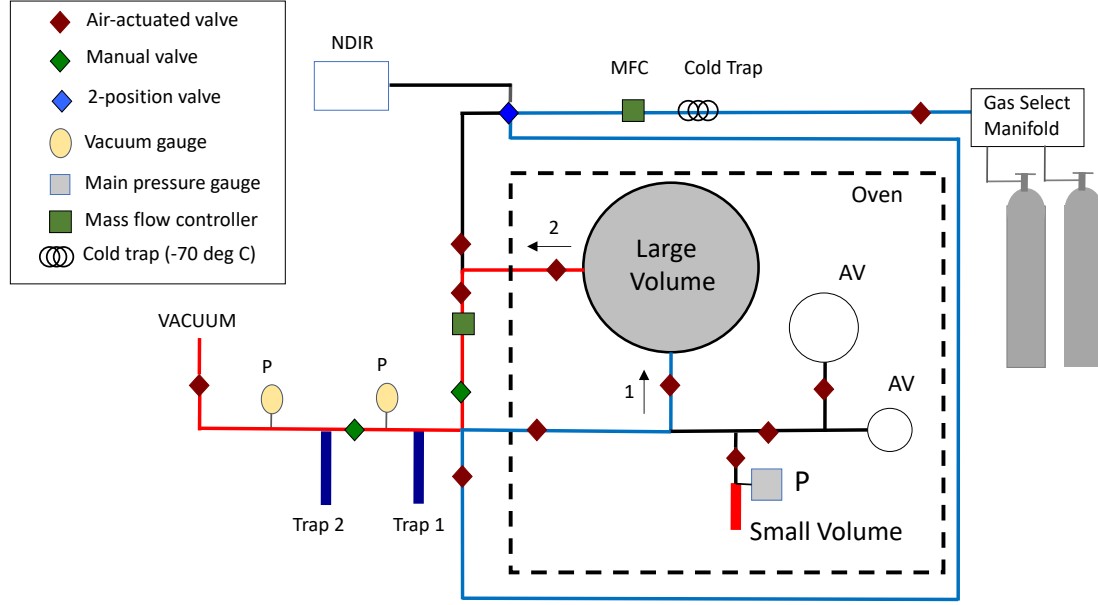

**Figure 1:** Schematic of the NOAA manometer. Air is passed through a cold trap (~ -69 °C) and mass flow controller into the large volume in the direction of arrow (1), shown as blue lines. After the temperature and pressure of the air in the large volume are recorded, the air is drawn from the large volume, in the direction of arrow (2) (red lines), and through traps 1 and 2 (approx. -197° C at 84 kPa) to cryogenically trap the $CO_2$. The $CO_2$ is cryogenically purified in glass traps 1 and 2, and then transferred




to the small volume where its pressure and temperature are determined. Auxiliary volumes ("AV") are used in separate experiments to determine the ratio of large and small volumes (volume ratio). The dashed line depicts a temperature-controlled oven housing the glass volumes and pressure gauge.

## 3 Reprocessing historical manometer data

Manometer data were obtained using software designed to read and store temperature and pressure data during a manometer run, and calculate the $CO_2$ mole fraction. Prior to each manometric episode, temperature and pressure were referenced to national standards (and to the SI) through calibration at accredited laboratories. Pressure and temperature calibration coefficients needed to convert measured variables to P and T, as well as the volume ratio, were hard-coded in this software. During the final P, T measurement, $CO_2$ was calculated periodically as the gas in the small volume warmed and equilibrated

to oven temperature during the final stage of measurement. An example of $CO_2$ mole fraction calculated as a function of time is shown in Fig. 2.

Mole fractions of $CO_2$ were previously determined as the maximum $X_{CO2}$ calculated during the final stage (Fig. 2), adjusted for $X_{N2O}$. There are two minor issues associated with this method that we correct with the implementation of the X2019 scale.

First, we recently discovered an error in the software used to calculate $X_{CO2}$. The second virial coefficient for $CO_2$ ($\beta_{CO2}$) (Sengers et al., 1971) was calculated corresponding to a temperature that was 10 K higher than the actual $T_{CO2}$ (320 K instead of 310 K) due to an interpolation error. Temperature was recorded correctly, but $\beta_{CO2}$ was calculated incorrectly. Consequently, $X_{CO2}$ was underestimated by about ~0.03 ppm at 400 ppm. Second, we recognize that the pressure in the small volume decreases slowly with time after the temperature of the small volume stabilizes (Fig. 2). For the 380 ppm sample shown in Fig. 2, the

rate of change in pressure is $-10^{-5}$ kPa s$^{-1}$, or $-0.036$ kPa hr$^{-1}$. We suspect that $CO_2$ absorbs to Viton O-rings and possibly adsorbs to surfaces of the small volume (Fig. 3). Separate tests conducted with pure $CO_2$ and Viton O-rings revealed $CO_2$ loss rates comparable to what is observed in the manometer small volume. Essential to the development of the X2019 scale was revisiting previous data and making corrections for the incorrect $\beta_{CO2}$ and the loss of $CO_2$ that occurred prior to the maximum measured $X_{CO2}$.


The results from all manometric determinations are stored in a database. Historical manometer results were adjusted using the following equation:

$$X_{CO2}\ (update) = X_{CO2}\ (original) + X_{virial\_correction} + X_{loss\_correction} \qquad (2)$$





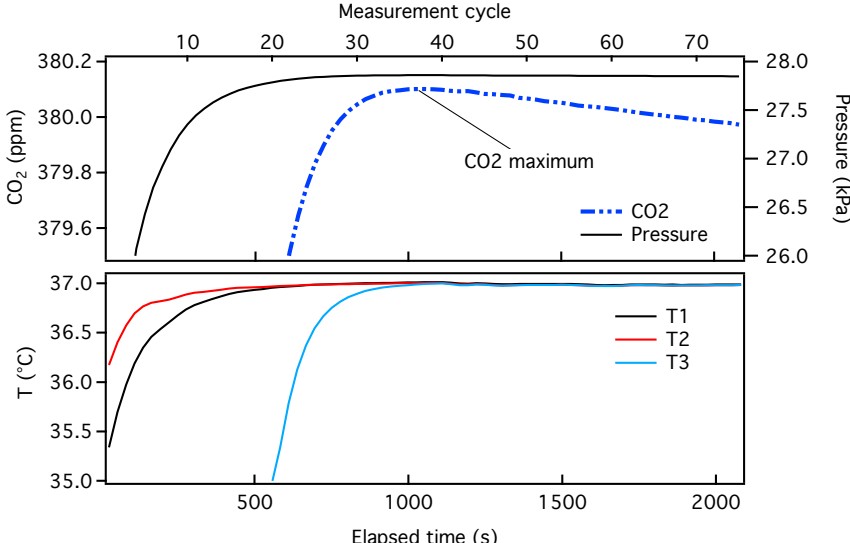

**Figure 2**: Typical data from a manometric run, showing the small volume pressure and $CO_2$ (+$N_2O$) calculated as a function of time (upper panel), and temperature measured at three locations within the oven (lower panel). Historical manometric records are time-stamped with "measurement cycle", which is shown on the upper x-axis. Here, each measurement cycle corresponds to ~ 30s. Temperature probe T3 is adjacent to the small volume and is cooled to liquid nitrogen temperature during extraction.

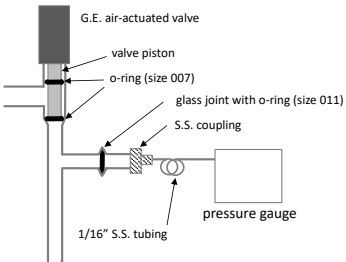

**Figure 3:** Diagram of the small volume showing position of Viton O-rings in the air-actuated valve (Glass Expansion, Pocasset, MA) and a glass joint.


## 3.1 Correcting for $\beta_{CO2}$

For $X_{virial\_correction}$, we first updated the data reduction software to calculate $\beta_{CO2}$ by correctly interpolating between the same $\beta_{CO2}$ coefficients used to define X2007 (-112.8 cm$^3$ mol$^{-1}$ at 310 K, and -104.8 cm$^3$ mol$^{-1}$ at 320 K (Zhao et al., 1997)). We then use the correct $\beta_{CO2}$ to calculate $X_{CO2}$ from pressure and temperature recorded in manometric data files, and compare to $X_{CO2}$ calculated using the original (incorrect) values for $\beta_{CO2}$. Fig. 4 shows differences between the original $X_{CO2}$ results ($\beta_{CO2}$ incorrect) and the updated results ($\beta_{CO2}$ correct). There are three representative periods that correspond to three nominally different volume ratios. The data show compact relationships with $CO_2$ mole fraction, as expected, since the mole fraction determined is largely a function of the pressure of $CO_2$ collected in the small volume. During each manometric determination, several temperatures were recorded. Since there are periods for which we do not know specifically which temperature records were used or the exact volume ratio used in the original calculation, we used three polynomial functions to estimate $X_{virial\_correction}$ corresponding to three time periods: 1996-1999, 1999-2003, and 2004-2016 (Fig. 4). The uncertainty associated with the estimated $X_{virial\_correction}$ is less than 0.01 ppm.

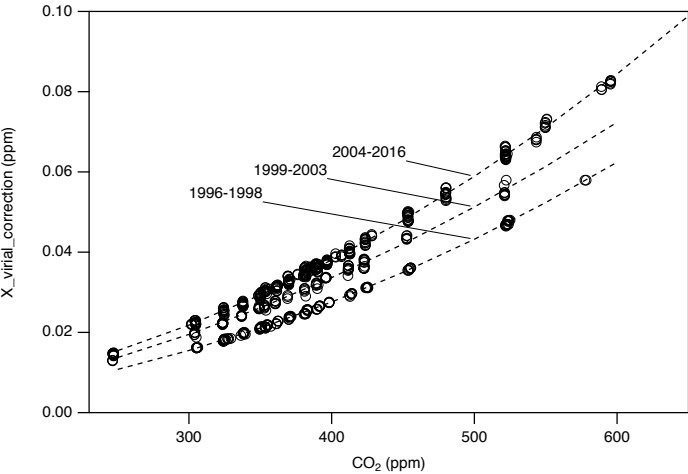

**Figure 4**: Corrections applied to account for an incorrect second virial coefficient ($X_{virial\_correction}$). Three second order polynomial functions were used corresponding to periods with nominally different volume ratios. Measurements performed after 2016 do not require this correction.

### 3.2 Correcting for $CO_2$ loss

To correct for $CO_2$ loss, we assume that loss of $CO_2$ to materials in the small volume begins soon after $CO_2$ sublimes and occurs at a constant rate. By extending the manometer run time out several hours, we can see that the loss rate decreases with time (see Supplemental Material). However, the loss rate is sufficiently linear over the short term that a linear correction is a reasonable approach.




We derive loss rates by fitting a linear function to the calculated $X_{CO2}$, beginning ~3 minutes after the maximum $CO_2$ and fitting 10-12 minutes of data (Fig. 5). This period corresponds to near-constant temperature and a steady decrease in pressure. After obtaining a loss rate from each data file, we correct the existing $CO_2$ record using the loss rate and elapsed time (expressed in terms of a measurement cycle, each approx. 30 s in duration).


$$X_{loss\_correction} = -a(t - t_0) \qquad (3)$$

were $a$ is the slope calculated from a record of $CO_2$ vs time as in Fig. 5 (ppm time$^{-1}$), $t$ is the time corresponding to the $CO_2$ maximum, and $t_0$ is the time at the start of the record, where we expect $CO_2$ loss to begin. Since $a<0$, $X_{loss\_correction}$ is positive.

As an example, the maximum $CO_2$ shown in Fig. 5 occurs at cycle 35 in the data file, ~1050 seconds after the liquid nitrogen was removed from the small volume. The slope ($a$), is $-0.0074 \pm 0.0002$ ppm min$^{-1}$. If the loss of $CO_2$ begins at time $t_0=0$, the correction required would be 0.13 ppm. After the liquid nitrogen is removed from the small volume, we estimate that the purified $CO_2$ reaches a temperature of 273 K within 1 minute, and 300 K within 3 minutes. Adsorption of $CO_2$ probably begins about 1 minute after the liquid nitrogen is removed. For many data records, we know that there was a delay of about 2 minutes

between the time the liquid nitrogen was removed and the first data record. While this cannot be confirmed for all records, this 2-minute delay is consistent with the rate of pressure rise observed, so we include this 2-minute delay in all data records. Thus, for elapsed time we use $t_{\_max\_CO2}$ + 2 minutes ($t_0$ = 2 min.). An error of 2 minutes in elapsed time would correspond to 0.015 ppm for a typical 400 ppm sample. Using an elapsed time of $t_{\_max\_CO2}$ + 2 min. (17.5 +2 min, or 39 measurement cycles) in the above example, the loss correction is 0.14 ppm.

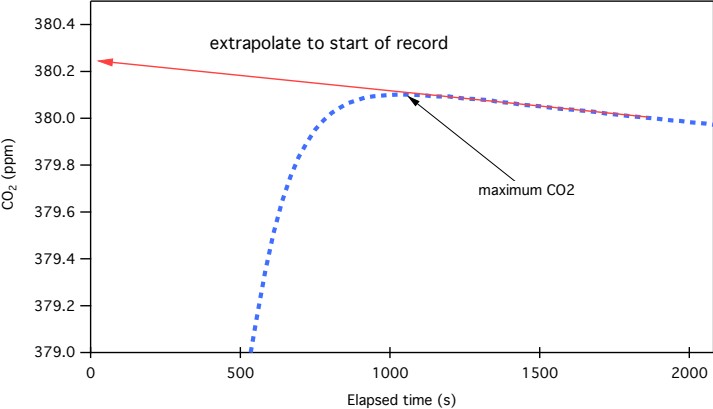


**Figure 5:** Typical manometer results showing $CO_2$ calculated as a function of time as the purified $CO_2$ warms while contained in the small volume. The loss rate is calculated from a linear fit as shown.

All loss rates and estimated uncertainties are shown in Fig. 6. There is some time dependence to the corrections applied,
possibly due to changes in materials (valves, O-rings, etc.). The rate of $CO_2$ loss has generally increased over time, however it
may have improved slightly after a new air-actuated valve with new Viton O-rings was installed in 2013.

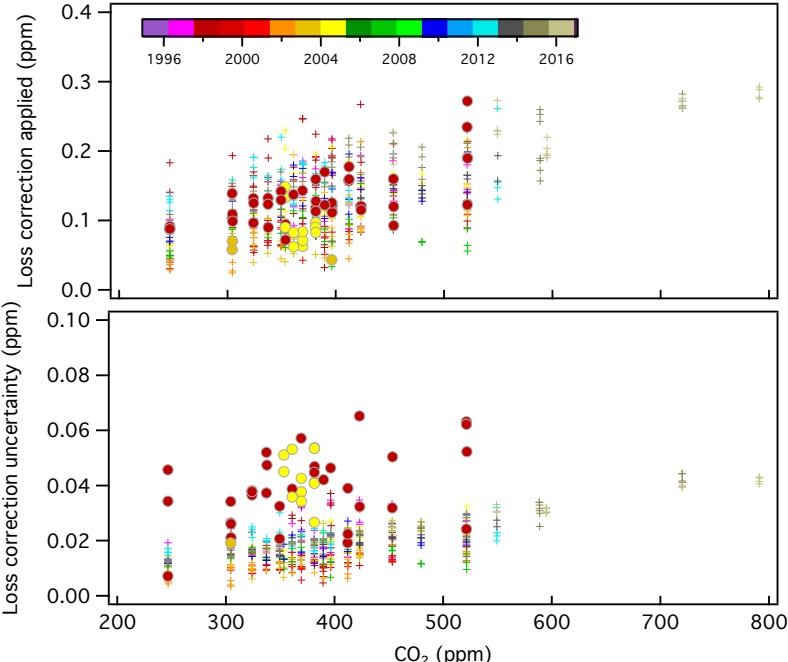

**Figure 6**: $CO_2$ loss correction ($X_{loss\_correction}$) applied to manometer results in developing the X2019 scale (upper panel), and
estimated uncertainty associated with the loss correction (lower panel), color-coded by year. The corrections are mole fraction
and time-dependent. Filled symbols correspond to runs in which two $CO_2$ maxima were observed (1998 and 2004), and loss
rates were determined from data after the second maxima. Forty-eight manometric runs were processed this way (8.9% of the
total).

## 4   Summary of manometer results

The X2007 scale was derived by averaging results from seven manometric episodes (1996, 1998, 2000, 2001, 2003, 2004,
2006) (Table 1). In developing X2019, we examined data files back to 1996, and applied the corrections discussed previously
(Fig. 7). There is not a 1:1 correspondence between original and reprocessed results. In a few cases, the original data appeared
abnormal and were flagged when developing X2019. In other cases, we were either unable to find raw files corresponding to
results in the database, or the records were not sufficient to calculate a $CO_2$ loss rate (data not stored for sufficient time). In
all, we were able to recover and apply corrections to 93% of the original data records.








**Figure 7:** History of manometric results, showing $CO_2$ (ppm) from the manometric database before corrections (black triangles), $CO_2$ values after applying $X_{virial}$ and $X_{loss\_correction}$ (red circles), and results from 2017 and 2020 for which the 2nd virial coefficient $\beta_{CO2}$ was calculated correctly and only the loss correction was applied (green diamonds). Note that cylinders

CC71578, CA08231, CB11054, and CC71605 have a shorter measurement history.



Higher variability in 1998 could be related to higher water vapor in samples extracted during that period. Manometric records from 1998 often did not show the characteristic single $CO_2$ maximum. Instead, those records show an initial "$CO_2$" peak, followed by a short decline, and then a secondary peak followed by the normal decline (see Supplemental Material, Fig. S1).

This secondary peak could be related to $H_2O$ desorbing from surfaces in the small volume. We have seen this pattern recently when the manometer has not been run for several weeks and tends to show characteristics of residual moisture (longer pump-down times and higher than normal $X_{CO2}$ results). For most of the records from 1998 and some records from 2004, $X_{loss\_correction}$ was determined from the time associated with the first peak in $CO_2$, and the loss rate determined after the second peak in $CO_2$. We used the later loss rates because it appears that the initial slopes (loss rates) are impacted by evolution of $H_2O$, and the loss

rates calculated after the second peak in $CO_2$ are more consistent with loss rates determined during other episodes. Although this introduces additional uncertainty, results from 1998 are generally consistent with those from other years (Fig. 7). Comparing 1998 results to other years, it would appear any potential impact of additional water vapor as an impurity is less than 0.1 ppm. Further, if we used the time associated with the second peak instead of that associated with the first peak, manometer results from 1998 and 2004 would be slightly greater, but this would translate into an increase of only 0.01 ppm in

the average manometric values for primary standards in the 250-520 ppm range.

It is also important to note that in May of 2014 we damaged the small volume during routine maintenance. New glassware and a new air-actuated valve (Glass Expansion, Pocasset, MA) were installed in August 2014. This meant that the volume ratio, which had been essentially constant since 2004, needed to be re-established. After establishing traceability for temperature and

pressure, we performed a number of volume ratio experiments and obtained a new volume ratio that was 2% larger than the previous one (see Supplemental Material). Results from the 2015 episode, with the new small volume and volume ratio, agree well with those from previous episodes. The mean difference between the 2012 episode and 2015 episode, for all primary standards in the 250-520 ppm range, is only 0.03 ppm.




**Table 1**: Primary standard $CO_2$ mole fractions (ppm) determined using the NOAA manometer. A lower case "x" is used here to indicate that these are mean values determined from manometric measurement, and have not yet been harmonized into a calibration scale. For x2007 we report the average manometric results from seven episodes (as the mean of the episode averages). For x2019 we averaged all valid recoverable data from 1996-2017 after correcting for $\beta_{CO2}$ and $CO_2$ loss. Note that primary ND17440 was put into service in 2010 to replace a standard that was thought to be drifting upward. ND17440 was not part of the original X2007 scale. CC71605 includes data from 2020.

| Cylinder | Avg. (x2007) | Avg. (x2019) | s.d. (x2019) | N (x2019) | $N_{ep}$ (x2019) |
|---|---|---|---|---|---|
| AL47-110 | 246.656 | 246.724 | 0.090 | 30 | 10 |
| AL47-102 | 304.370 | 304.495 | 0.099 | 31 | 9 |
| AL47-111 | 324.004 | 324.134 | 0.116 | 38 | 10 |
| AL47-130 | 337.271 | 337.403 | 0.087 | 31 | 10 |
| AL47-121 | 349.387 | 349.515 | 0.089 | 28 | 9 |
| AL47-139 | 360.905 | 361.054 | 0.056 | 30 | 10 |
| AL47-105 | 369.378 | 369.523 | 0.104 | 33 | 10 |
| AL47-136 | 381.335 | 381.487 | 0.092 | 34 | 10 |
| AL47-146 | 389.569 | 389.731 | 0.100 | 35 | 10 |
| AL47-101 | 396.333 | 396.495 | 0.130 | 34 | 10 |
| AL47-106 | 412.069 | 412.231 | 0.103 | 34 | 10 |
| AL47-123 | 423.086 | 423.218 | 0.112 | 32 | 10 |
| AL47-107 | 453.078 | 453.255 | 0.144 | 37 | 10 |
| ND17440 | 479.510 | 479.720 | 0.054 | 15 | 5 |
| AL47-132 | 521.410 | 521.605 | 0.122 | 41 | 10 |
| CC71578 | not used | 549.571 | 0.091 | 15 | 4 |
| CA08231 | not used | 588.909 | 0.090 | 12 | 3 |
| CB11054 | not used | 720.288 | 0.126 | 11 | 3 |
| CC71605 | not used | 791.551 | 0.160 | 13 | 3 |

N = total number of measurements, $N_{ep}$ = number of episodes

## 5 Drift assessment

The mole fraction of $CO_2$ (in air) in aluminum cylinders can increase with use (Langenfelds et al., 2005; Leuenberger et al., 2015; Schibig et al., 2018). Our experience suggests that $X_{CO2}$ is relatively stable over the useful life of a cylinder when used sparingly at flowrates ~0.3 L min$^{-1}$ or lower, but can increase as the pressure drops below about 15% of the fill pressure. However, it is worth noting that detecting small drift rates over decades is very difficult because it requires a stable reference with comparably low uncertainties. At the end of the 2015 measurement episode, all 15 primary standards contained at least

one third of the original gas, with pressures of at least 4.4 MPa (600 psi), and most contained more than 6 MPa.





Drift in the X2007 scale was assessed through repeated manometric measurement. Only AL47-103 (no longer in use) was found to be drifting. With the update to X2019, we applied corrections to the primary standards that were both a function of mole fraction and time. We therefore need to reassess the possibility of drift in the primary standards. We performed a weighted least squares linear fit to the mean mole fraction determined during each episode. Uncertainties were estimated by combining the manometer repeatability during each episode ($\sigma_i/\sqrt{N_i}$), where $\sigma_i$ is the standard deviation of results within episode "i", and $N_i$ is the number of measurements during that episode, with the relative uncertainty in the volume ratio and the average uncertainty associated with $X_{virial\_correction}$ and $X_{loss\_correction}$ for each episode (0.02-0.04 ppm). We lack sufficient information to fully evaluate the uncertainty in the volume ratio dating back to the earliest periods, so we assume that our current uncertainty assessment is valid for the entire record. We consider each episode independent since traceability to national standards for temperature and pressure was established prior to each episode, and do not include uncertainty components common to all episodes (which include components of the volume ratio uncertainty related to temperature gradients in the oven, and differences in volume ratio obtained using difference gases ($N_2$, air, and argon)). We estimate the total uncertainty in the volume ratio to be 0.016% (see Supplemental Material). Excluding components common to all episodes, we use 0.012% for uncertainty on the volume ratio in the drift assessment.

Drift rates, in ppm per decade, are summarized in Fig. 8 (see also Table S1). For primary standards with $X_{CO2}$ > 530 ppm, the manometric histories are too short to adequately assess drift. For those with $X_{CO2}$ in the range 250-520 ppm, all but two show positive drift, although none is significant at the 95% C.L. While some calculated drift rates are of order 0.05 ppm/decade, we are unable to detect drift rates less than 0.1 ppm/decade owing mostly to the uncertainty associated with the volume ratio and repeatability. The average drift rate among standards in the 350-450 ppm range is 0.019 ppm decade$^{-1}$, which would have only a minor impact on the heart of the X2019 scale if drift rates shown in Fig. 8 were incorporated. Thus, while relative drift among cylinders can be observed over short time periods, as in Leuenberger et al. (2015) and Schibig et al. (2018), detecting long-term drift on an absolute basis is difficult. Still, drift in cylinders is typically small compared to the growth rate of atmospheric $CO_2$ (~ 2 ppm yr$^{-1}$).

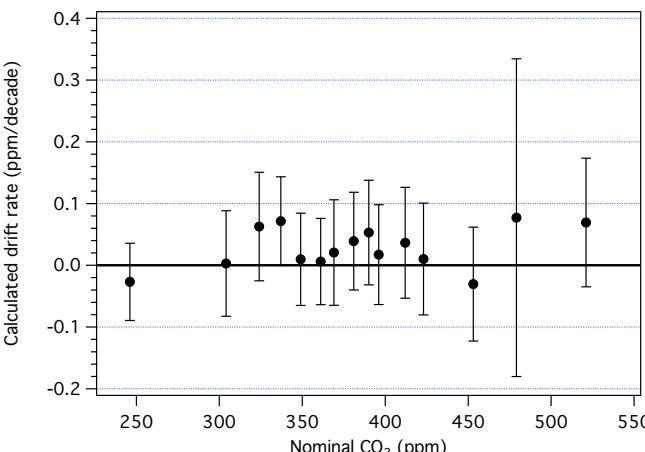

**Figure 8:** Drift rates (ppm decade$^{-1}$) determined from 5-10 manometric episodes. Error bars are 95% confidence limits.

## 6 Defining the X2019 WMO CO$_2$ mole fraction scale

Primary standards were analyzed using the laser-spectroscopy system described in Tans et al. (2017). These data were then
used to harmonize the standards and define a scale. Each primary standard was analyzed six times relative to a ~400 ppm
reference cylinder. On this analysis system we treat the three major isotopologues of CO$_2$ separately to eliminate subtle biases
due to variations in isotopic compositions among the standards and between samples and references cylinders. We harmonized
the primary standards using only the major ($^{16}O^{12}C^{16}O$) isotopologue measurement.

The average manometer results are decomposed into the component mole fractions of $^{16}O^{12}C^{16}O$, $^{16}O^{13}C^{16}O$, and $^{16}O^{12}C^{18}O$
based on the $\delta^{13}C$ and $\delta^{18}O$ assignments for each standard (Tans et al., 2017). The isotopic assignments for the primary
standards were made by filling a pair of flask samples from each primary standard and having the flasks measured by the
Stable Isotope Laboratory at the University of Colorado, Institute of Arctic and Alpine Research (INSTAAR) relative to the
JRAS-06 realization of VPDB-CO$_2$ (Sylvia Michel, *personal communication*). Typical analytical uncertainties reported for
flask measurements by INSTAAR are ±0.014 ‰ and ±0.035 ‰ for $\delta^{13}C$ and $\delta^{18}O$ respectively (White et al., 2015).
Additionally, there is uncertainty in the tie to the JRAS-06 scale realization which is currently being evaluated as part of the
conversion of INSTAAR data to the JRAS-06 scale realization. Based on a re-evaluation of recent comparisons with other
laboratories, it is expected to be less than 0.05 ‰ for both $\delta^{13}C$ and $\delta^{18}O$ (Sylvia Michel, *personal communication*). These
uncertainties are insignificant relative to the uncertainty in the manometric determination of total CO$_2$ in terms of the calculated
mole fraction of the $^{16}O^{12}C^{16}O$ isotopologue.

The uncertainties on flask measurements at INSTAAR listed above are determined for ambient atmospheric samples (~-7.5 to
-9 ‰). Several of the primary standards are depleted relative to the atmosphere (see Table 2) and this could increase the





uncertainty of these measurements due to scale contraction in the measurements at INSTAAR. At $\delta^{13}C$ =-20‰ and $\delta^{18}O$ =-
20‰, Wendeberg et al. (2013) found the INSTAAR realization of VPDB-$CO_2$ to be offset from JRAS-06 by approximately
0.2‰ in $\delta^{13}C$ and 0.8‰ in $\delta^{18}O$. This was primarily due to scale contraction due to the instrumentation in use at INSTAAR.
Subsequent conversion of the INSTAAR records to JRAS-06 is not expected to correct for the scale contraction in historical
measurements since these measurements were not done with two-point normalization. Errors in the isotopic assignments of
the primary standards of this magnitude due to scale contraction issues will result in errors of less than 0.01 ppm in the
calculated $^{16}O^{12}C^{16}O$ mole fraction. We therefore feel confident that we can harmonize the primary standards based on the
$^{16}O^{12}C^{16}O$ measurements only.

A linear fit (orthogonal distance regression) was applied to the normalized analyzer response and the $^{16}O^{12}C^{16}O$ component of
the average manometer results. This was repeated six times over three years. To test the sensitivity of the harmonization
process, we performed an orthogonal distance regression with two variations of manometric average values and two variations
of weighting factors for each primary standard (four combinations). For the manometric data, we used either the average of all
manometric measurements of each primary standard, or the weighted average from each measurement episode. For the weights
in the regression, we used either the inverse variance ($1/\sigma^2$) (as in Table 1) or the square of the inverse standard error. All four
variations give essentially the same result (within 0.01 ppm near 400 ppm). Therefore, the X2019 scale is defined from an
orthogonal distance linear regression using the average manometric result and standard deviation (using $1/\sigma^2$ as weighting
factors) for each cylinder ("Avg. (x2019)" and "s.d. (x2019)" in Table 1).

Fig. 9 shows the residuals from six analysis periods over three years associated with harmonization. There is good agreement
among the different analysis periods, indicating that variability seen in the residuals relates to the manometer average values.
For each primary standard, we corrected the $CO_2$ mole fraction by the mean residual from the linear fit (Table 2). The X2019
scale is defined as the average residual-corrected mole fraction, determined over six analysis periods, for each primary
standard. In this way, the scale is defined over a range, with better consistency and smaller uncertainty compared to individual
primary standards. For X2019, we include the 15 primary standards used to define the X2007 scale, plus four additional
primary standards with $X_{CO2} > 530$ ppm. Additional primary standards in the upper range help to constrain the fit and reduce
end-effects. Many residuals are less than 0.05 ppm, but the newer standards in the upper $CO_2$ range show larger residuals.
Some of this may be due to their short measurement history compared to standards in the 250-520 ppm range. Finally, while
harmonization is not strictly necessary if all primary standards are to be analyzed at the same time when propagating the scale,
it provides some insurance on the potential loss of a primary standard. By assigning mole fractions consistent with the best fit
response, loss of one or two standards from the suite of 19, especially in the middle of the $X_{CO2}$, range would not be catastrophic.

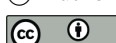



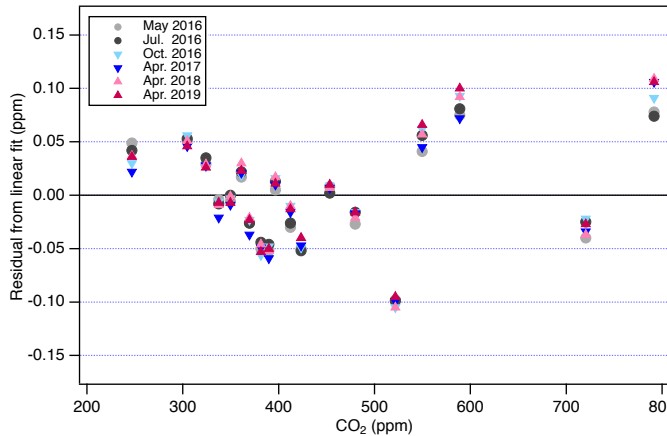


**Figure 9:** Residuals from a linear fit to analysis data obtained on six different days. The standard deviation of all residuals is 0.05 ppm.

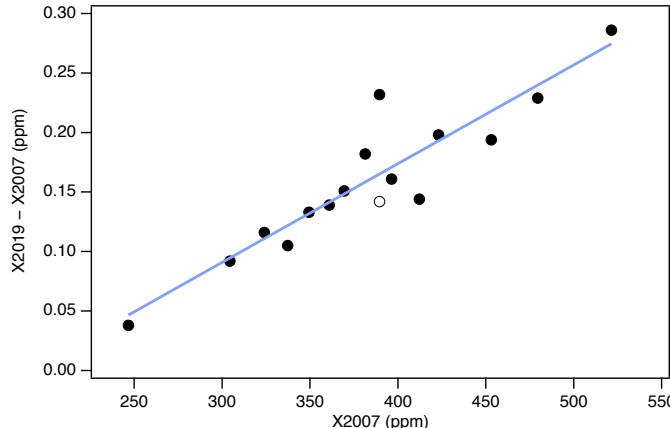

**Figure 10**: Differences between X2019 and X2007 assignments for the 15 primary standards used to define scale X2007 (black

symbols) and the best-fit line (blue line). The open symbol corresponds to primary AL47-146 with value 389.64 ppm (we used 389.55 ppm in 2007 by mistake).

Fig. 10 shows differences between primary standard assignments on X2019 and X2007. As expected, the differences are a function of mole fraction, since both the virial correction and loss correction are functions of mole fraction. The scale difference

based on primary standards alone (not including scale transfer) is 0.17 ppm at 400 ppm, and the average scale correction over the range 250-520 ppm is 0.04%. Some of the scatter in Fig. 10 is due to updated assignments owing to a longer measurement record for X2019 compared to X2007. However, the largest deviation is due to a mis-assigned value: the assigned value for AL47-146 was inadvertently listed as 389.55 in our database instead of 389.64. The implications of this mis-assignment are discussed in section 9.





**Table 2**: WMO primary standard assignments on X2007 and X2019 scales. Assignments were determined following analysis and residual correction by NDIR (X2007) and laser-spectroscopy (X2019). The average ratio of primary standards on scales X2019/X2007 is 1.00040, with standard deviation 0.00011.

| Cylinder | Assigned $X_{CO2}$ X2007 (ppm) | $\delta^{13}C$ (‰) | $\delta^{18}O$ (‰) | Assigned $X_{CO2}$ X2019 (ppm) | X2019 minus X2007 (ppm) | X2019/X2007 |
|---|---|---|---|---|---|---|
| AL47-110 | 246.65 | -6.8 | 0.1 | 246.69 | 0.038 | 1.00015 |
| AL47-102 | 304.35 | -7.5 | 0.2 | 304.45 | 0.092 | 1.00030 |
| AL47-111 | 323.99 | -8.0 | -0.3 | 324.11 | 0.116 | 1.00036 |
| AL47-130 | 337.31 | -7.6 | -0.5 | 337.41 | 0.105 | 1.00031 |
| AL47-121 | 349.39 | -7.7 | 0.1 | 349.52 | 0.133 | 1.00038 |
| AL47-103* | 353.24 | | | *not used* | | |
| AL47-139 | 360.89 | -8.5 | -1.6 | 361.03 | 0.139 | 1.00039 |
| AL47-105 | 369.40 | -9.2 | -1.8 | 369.55 | 0.151 | 1.00041 |
| AL47-136 | 381.36 | -10.1 | -3.0 | 381.54 | 0.182 | 1.00048 |
| AL47-146 | 389.55 | -10.7 | -4.0 | 389.78 | 0.232 | 1.00060 |
| AL47-101 | 396.32 | -11.2 | -4.5 | 396.48 | 0.161 | 1.00041 |
| AL47-106 | 412.11 | -12.2 | -5.7 | 412.26 | 0.144 | 1.00035 |
| AL47-123 | 423.07 | -12.9 | -6.8 | 423.26 | 0.198 | 1.00047 |
| AL47-107 | 453.05 | -14.6 | -9.6 | 453.25 | 0.194 | 1.00043 |
| ND17440* | 479.51 | -14.0 | -13.8 | 479.74 | 0.229 | 1.00048 |
| AL47-132 | 521.42 | -17.7 | -14.8 | 521.71 | 0.286 | 1.00055 |
| CC71578 | *not used* | -15.2 | -12.7 | 549.52 | | |
| CA08231 | *not used* | -8.9 | -13.8 | 588.82 | | |
| CB11054 | *not used* | -8.5 | -19.2 | 720.32 | | |
| CC71605 | *not used* | -8.8 | -21.1 | 791.46 | | |

\* AL47-103 was found to be drifting and was replaced with ND17440 in 2010.

## 7 Independent Assessment

Revision of the X2007 scale relies on the assumption that the loss of $CO_2$ to Viton O-rings in the small volume of the manometer can be adequately addressed by linear extrapolation (Fig 5). Knowledge on $CO_2$ losses prior to the availability of representative pressure and temperature measurements (during the time while the small volume is warming) is lacking. Experiments in which pure $CO_2$ was loaded into the small volume by overpressure (not transfer by cryogenic extraction) suggest that the loss process is initially non-linear and approaches a linear rate after about 10 minutes. If this is true, then the correction we apply is too small (by ~0.2 ppm) (see Supplemental Material). However, these experiments were not carried out under the same conditions used to extract $CO_2$ from air, so we cannot be sure that they are representative. Therefore, we



explored an independent method to provide insight into potential bias in the X2007 scale and our attempt to correct for that bias.

### 7.1 Comparison to in-house, gravimetrically-prepared standards

We prepared $CO_2$ primary standards using a gravimetric method (Hall et al., 2019). Briefly, known masses of highly pure $CO_2$ were introduced into 29.5-L aluminum cylinders and diluted with known masses of $CO_2$-free air. Uncertainties were reduced by preparing standards in one step and by accounting for $CO_2$ likely to be adsorbed to cylinder walls at high pressure (Schibig et al., 2018). These standards were analyzed by laser spectroscopy and assigned $X_{CO2}$ values on the X2019 scale (Table 3). The X2019 assignments are consistent with the gravimetrically-prepared values, with an average difference of 0.03 ppm, and an

average ratio of 1.00008 (Table 3). If the gravimetric standards were used to define a calibration scale, it would, on average, be 0.045% greater than the X2007 scale (avg ratio 1.00045, std. dev. 0.00017) (Hall et al., 2019). This is very close to the average ratio of 1.00040 derived by correcting historical manometric data (Table 2).

**Table 3:** Comparison of gravimetrically-prepared standards to the X2019 scale (prep. = prepared value; unc. = standard
uncertainty, ~68% confidence level). Gravimetric standards were analyzed by laser-spectroscopy.

| cylinder | Grav. prep. (ppm) | unc. (ppm) | X2019 (ppm) | unc. (ppm) | Difference X2019-prep (ppm) | unc. (ppm) | Ratio X2019/prep |
|---|---|---|---|---|---|---|---|
| CB11873 | 357.55 | 0.06 | 357.56 | 0.08 | 0.01 | 0.10 | 1.00004 |
| CB11906 | 397.50 | 0.06 | 397.54 | 0.09 | 0.04 | 0.11 | 1.00011 |
| CB11941 | 405.34 | 0.07 | 405.46 | 0.09 | 0.12 | 0.12 | 1.00030 |
| CB11976 | 449.30 | 0.08 | 449.27 | 0.10 | -0.03 | 0.12 | 0.99993 |
| CB12009 | 491.76 | 0.08 | 491.76 | 0.10 | 0.00 | 0.13 | 0.99999 |
| | | | | | avg, 0.03 | | avg. 1.00008 |

### 7.2 Comparison with NIST

Based on an exchange of 30 tertiary standards in 2010, NOAA and the National Institute of Standards and Technology (NIST) reported an average difference of 0.19 ± 0.03 ppm (NOAA lower) over the range 388-394 ppm (Rhoderick et al., 2016). After adjusting NOAA results to X2019, differences range from −0.08 to +0.07 ppm, with a mean difference of 0.0 ± 0.03 ppm.

### 7.3 Key Comparison CCQM-K120a

NOAA recently participated in an international comparison (CCQM-K120a) organized under the auspices of the Consultative Committee for Amount of Substance: Metrology in Chemistry and Biology (CCQM), and hosted by the BIPM. Fourteen National Metrology Institutes or Designated Institutes submitted compressed gas standards for analysis by FTIR and GC-FID (Flores et al., 2018). Key Comparison Reference Values (KCRVs) were calculated using FTIR results from consistent sets of



standards submitted by participants near 380 and 480 ppm. NOAA standards were value-assigned at GML on a provisional

version of the X2019 scale, identified as X2017p, which is within 0.02 ppm of the now complete X2019 scale. During K120a,

NOAA standards differed from the reference values by -0.01 ± 0.23 ppm at 380 ppm, and by -0.10 ± 0.28 ppm at 480 ppm

(uncertainties ~95% confidence level). Had we submitted values on X2007, the NOAA samples would have been

approximately 0.16 ppm lower than the reference value at 380 ppm, and 0.24 ppm lower at 480 ppm. While the X2007 scale

would also likely have agreed with the reference values within uncertainties at 380 ppm and 480 ppm during K120a, better

agreement was achieved with X2017p, and hence also with X2019.

## 8  Uncertainty Analysis

Here, we estimate the total uncertainty associated with a $CO_2$ determination on the X2019 scale. We extend the work of (Zhao

and Tans, 2006), following accepted methods for uncertainty propagation (JCGM, 2008). To arrive at an uncertainty estimate,

we use equation (4), which is a modified version of equation (1), and propagate uncertainties over a range of $CO_2$ mole

fractions. We include the terms $X_{virial\_correction}$ and $X_{loss\_correction}$ since the X2019 scale was derived based on these corrections.

Future manometric analysis will not include the term $X_{virial\_correction}$ since $\beta_{CO2}$ is now correctly determined. We also include the

term $X_{H2O}$ and estimated uncertainty even though we do not correct for water vapor in the final sample ($X_{H2O} = 0$).

$$X_{CO2} = (\phi^{-1})\frac{P_{CO2}T_{air}}{P_{air}T_{CO2}}(1 + A_1 - A_2) - X_{N2O} - X_{H2O} + X_{virial_{correction}} + X_{loss_{correction}} \quad (4)$$

### 8.1 Purity Assessment

The primary function of the separation steps is to remove $H_2O$ from the extracted $CO_2$. The purified $CO_2$ introduced into the

small volume contains $N_2O$ and trace amounts of other gases. Considering the major constituents in air and their boiling points

under the conditions at which $CO_2$ is trapped, Zhao et al. (1997) concluded that a correction for $N_2O$ sufficient to account for

impurities in the purified $CO_2$. We go one step further here by verifying, through analysis, that the purified $CO_2$ is, indeed,

highly pure and that additional purity corrections are not needed.

We used the manometer to extract $CO_2$ from a 380 ppm air sample. At the end of a normal manometer run, we transferred the

purified $CO_2$, first to a stainless steel tube (5 mL volume with stainless steel metal bellows valve) and then to a 2.3 L stainless

steel flask with a stainless steel metal bellows valve. We then added ~0.24 MPa (35 psia) UHP-grade nitrogen to create a

mixture with $X_{CO2}$ at approximately 380 ppm, the same as that of the original air. We analyzed this mixture by GC-MS, GC-

FID (Dlugokencky et al., 2005) and GC-ECD (Hall et al., 2011). We confirmed that gases likely to be trapped in the extraction

step (nitrous oxide, ethane, propane, some chlorofluorocarbons) are present in the purified $CO_2$ sample. The combined mole





fraction of all gases measured in the flask, excluding $CO_2$, $N_2O$, and $H_2O$, was 6 parts per billion (ppb). Of this 6 ppb, we found ~3.4 ppb Xe, 0.8 ppb ethane, 0.5 ppb $CCl_2F_2$ (CFC-12), 0.2 ppb $CFCl_3$ (CFC-11) and trace amounts of other halogenated gases. Had Xe been quantitatively trapped and retained, we would have found ~87 ppb, which would then require a correction. $CH_4$ was not detected in the purified $CO_2$ sample, confirming that $CH_4$ is not trapped during the extraction process.

We did not attempt to measure $H_2O$, krypton, argon, and oxygen since these would either not be trapped at the pressure of the traps (~4 kPa), or would likely be present at very low levels and would be difficult to measure. The water vapor content of our primary standards is < 2 ppm, and after two cryogenic separation steps we expect $H_2O$ to be <0.03 ppm. While we do not make a correction for water vapor that might remain in the final sample, we do include an estimate in the uncertainty budget. With traps 1 and 2 at -67 °C, we estimate that 98% of the water vapor in the sample would be removed in trap 1, and 50% of the

remaining water vapor would be removed in trap 2. This would correspond to 0.02 ppm in the measured $CO_2$. Because the trap temperatures vary from run to run (-65°C to -70°C), we include an uncertainty of 0.03 ppm in the uncertainty propagation.

## 8.2 Uncertainty Estimate

We establish traceability of manometric measurements to national temperature and pressure standards. Prior to a measurement episode, three platinum resistance thermometers, one thermistor, and a piston gauge are typically sent to an accredited

laboratory for calibration (National Voluntary Laboratory Accreditation Program, NVLAP). We estimate the uncertainties associated with measurement of temperature and pressure from uncertainties reported by the calibration laboratories, repeatability during a manometer run, and experience. We use these to estimate the uncertainty in the volume ratio. Estimates for standard uncertainties (1-sigma) are described in the Supplemental Material. We calculated standard uncertainties for $X_{CO2}$ using software developed by NIST (*metRology*) (http://cran.at.r-project.org/web/packages/metRology/). The *metRology*

software allows the user to input a measurement equation, central values, standard uncertainties, and estimated degrees of freedom for each variable. The software then calculates a central result ($X_{CO2}$ in this case) and a standard uncertainty following methods described in JCGM (2008). We multiply the standard uncertainty by 2 to arrive at an expanded uncertainty (~95% confidence level). Uncertainty components listed in Table S4 are similar to those estimated by Zhao and Tans (2006) except for the uncertainty associated with the volume ratio. We calculate a larger uncertainty for Φ, in part, because we observed

small temperature gradients in the oven, and hence our ability to measure the gas temperature at each stage of the expansion sequence with existing equipment is probably less certain than previously estimated (Zhao and Tans, 2006).

By calculating expanded uncertainties over a range of mole fractions, we arrive at a general expression for the expanded uncertainty ($\mu X_{CO2}$) as a function $X_{CO2}$:

$$\mu X_{CO2} = 0.000287 * X_{CO2} + 0.074 \quad (ppm) \tag{5}$$

Using equation (5), the expanded uncertainty at 400 ppm is 0.19 ppm, or 0.047%. This estimate is only slightly larger than that estimated by Zhao and Tans (2006) (2*0.069 = 0.14 ppm).

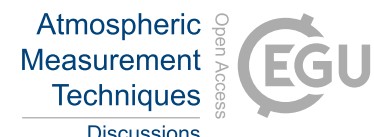

## 9 Scale Implementation

As discussed above, the implementation of the scale involves the harmonization of primary standard manometric results
through analysis, with assigned mole fractions derived using a linear response function based on spectroscopic analysis. These
assigned mole fractions are then used to define the X2019 scale, and transfer that scale to lower-order standards.

In the hierarchy of value-assignment, standards used to support NOAA atmospheric measurements and those distributed by
the CCL are known as "tertiary standards". Recalculating tertiary standard values on the X2019 scale involves three steps: 1)
updating primary standards to X2019, 2) re-assigning secondary standards based on primary-secondary comparisons (note that
some secondaries were re-assigned based on additional data not available upon initial assignment), 3) and re-assigning tertiary
standards based on updated daily response functions, relative to secondaries. Here we present the impact of the X2019 scale
update on tertiary value assignments dating back to 1995. In a subsequent section we present the implications of the scale
update on NOAA atmospheric measurements.


Tertiary standards are value-assigned based on analysis vs secondary standards (Zhao and Tans, 2006). From 1995 to October
2016, value assignment was performed by NDIR (Siemens Ultramat-3, -5, or -6F; LiCor Li-6251, Li-6252, or Li-7000), and
from November 2016 by laser spectroscopy (Picarro G2301; Los Gatos Research CCIA-46-EP; Aerodyne Research Inc. QC-
TILDAS-CS). There was an approximately 12-month overlap period where tertiary standards were run on both systems. The
NDIR response to $CO_2$ is typically non-linear. For analysis on a given day, a quadratic response function was determined based
on four secondary standards which were previously value-assigned based on similar mole fraction dependent subsets of the
suite of primary standards. Secondary standards were selected such that $X_{CO2}$ spanned the range of tertiary standards to be
calibrated. For example, analysis of a nominal 380 ppm tertiary standard would typically involve secondary standards at 370,
380, 390, and 400 ppm (10 ppm spacing). For $X_{CO2}$ greater than 450 ppm, three secondaries, spaced ~25 ppm apart, were used.
For analysis by laser-spectroscopy, 16 secondary standards over the range 250-800 ppm (prior to April 2020, 14 secondary
standards covering 250 - 600 ppm) are used to define response curves for the three major isotopologues of $CO_2$ ($^{16}O^{12}C^{16}O$,
$^{16}O^{13}C^{16}O$, and $^{16}O^{12}C^{18}O$). The mole fraction of each of the three major isotopologues is measured and then converted into
total $CO_2$, $\delta^{13}C$, and $\delta^{18}O$, accounting for the unmeasured minor isotopologues as described in Tans $et$ $al$. (2017).

Upon revision to X2019, all secondary standards used as far back as 1979 were re-evaluated. Secondary standards were
compared to primary standards multiple times during their use. A statistical test and expert judgement were employed to
evaluate drift in secondary standards. The statistical test was occasionally overruled in cases where we suspect a step change
due to change in instrumentation was the underlying driver rather than drift in the secondary standard. If drift was suspected,
a weighted linear or polynomial function was fit to the data (weighted by instrument reproducibility, 0.03 and 0.01 ppm for
the NDIR and laser spectroscopic systems respectively) and a time-dependent mole fraction used. Note that it is easier to detect



drift in secondary standards compared to primary standards because we evaluate secondary standards relative to the scale defined by many standards. Thus, the limiting factor is measurement reproducibility and not the absolute uncertainty of the scale.

During this re-evaluation, the drift status of some secondary standards was updated, with more data being available compared to when drift rates were first assigned. Thus, some standards that had previously assigned time-dependent values are now held constant, and vice-versa. Generally, the X2019 scale is more consistent across mole fraction and time, and therefore the new evaluations for secondary standard drift are considered more reliable. After updating secondary standard value assignments to X2019, $X_{CO2}$ for all tertiary standards dating to 1979 were re-assigned from raw data. We focus here mainly on the period from
1995 onward because our role as a WMO/GAW CCL began in 1995.

Fig. 11 shows differences between tertiary standard assignments on X2019 and X2007, from 1995 through February 2020. The overall scale difference is clearly a function of mole fraction, with the difference approximately 0.18 ppm at 400 ppm. It is immediately obvious that differences are not a perfect linear function of mole fraction. Differences that are consistent over
several months can be seen as coherent traces in Fig. 11. The coherent differences are due to secondaries being exhausted and replaced by others at slightly different mole fractions. Even though tertiary standards were bracketed by secondaries during analysis, limitations in the ability to value-assign any particular secondary standard, coupled with the limitations associated with fitting a quadratic response function to three or four secondaries contributes to variability. Even so, most of the year-to-year variability at a particular mole fraction is less than 0.02 ppm (1-sigma). Outliers, such as those corresponding to analysis
performed in the mid-1990s above 400 ppm (red and purple symbols), are the result of extrapolation beyond the range of the secondaries. Prior to 1997, the highest secondary standard in regular use was 390 ppm.

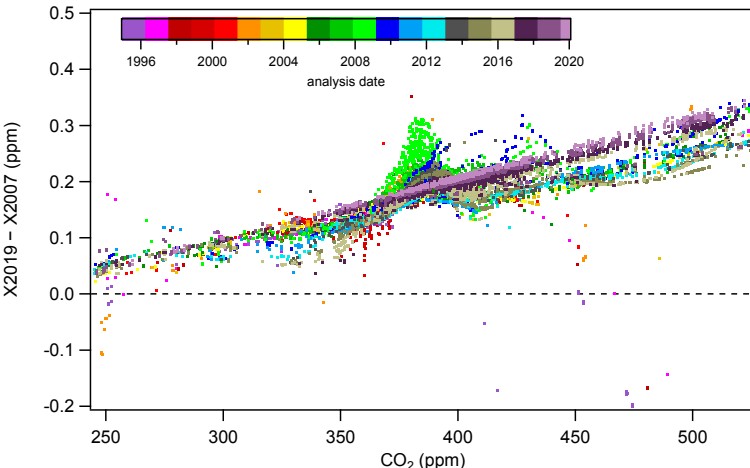





**Figure 11**: Differences between X2019 and X2007 assignments to tertiary standards from 1995 to 2020. Each data point represents one analysis records (over 25,000 records shown), and a full calibration of a tertiary standard involves multiple

analysis records.

The more prominent variations evident in Fig. 11 stem from re-assignment of primary and secondary standards, the non-linear response of NDIR instruments, and the nature of the value-assignment process. Scale differences appear significantly larger during 2008-2009 over the 360-390 ppm range (light green symbols). These value assignments, which involved around 600

analysis records (less than 3% of the total number), are inconsistent with most other data due to a revision of $X_{CO2}$ assigned to a particular secondary standard (CA01982) in use at the time. This particular secondary was assigned a value of 391.87 on the X2007 scale in 2008 when compared to primary standards. However, incorporating subsequent analysis of this cylinder against primary standards, it was evident that the cylinder was drifting upward rapidly. This secondary standard drifted ~0.2 ppm in two years (not common), but that drift was not accounted for in the X2007 value assignment, which caused the value used for

data reduction to be too low. The drift is accounted for in the X2019 value assignment leading to larger X2019-X2007 differences for tertiary standards measured against this secondary standard.

The more recent data based on analysis by laser spectroscopy are represented as dark purple and maroon colors in Fig. 11. These show a more linear relationship without the wavy structure, as expected for an instrument with a linear response

calibrated over the entire scale range. The fact that the laser spectroscopic results do not agree with the NDIR data in the upper $X_{CO2}$ range (> 420 ppm) is due the use of secondary standards on this system that were not well-characterized. Value-assignments for these secondary standards were determined on the NDIR system and thus incorporate the biases associated with that system on X2007. They were not well characterized when they went into service, especially at the upper end of the range where we effectively expanded the calibration range in anticipation of the X2019 revision. We now have more

information on these secondary standards, including analysis vs the primary standards on the laser spectroscopic system and can better define them on X2019.

It is important to note that differences in value-assignment between the NDIR and laser spectroscopic system (Fig. 11) are only present on the X2007 scale. The X2019 revision resolves the underlying cause of the offsets. Fig. 12 shows the results

from the ~12 month overlap during which tertiary standards were analyzed on both systems. There is a clear mole fraction dependence to the offset on the X2007 scale that is due to the assigned values of the primary standards and the method used for scale transfer using the NDIR. The X2007 primary standard assignments (Table 2), based on harmonization by NDIR analysis, were not as robust as we thought. The X2007 scale was based on relatively few NDIR analysis runs, and as such the residuals were not as well defined as they are for X2019 (Fig. 9). By using small subsets of standards to calibrate the NDIR,

the data reduction of the NDIR system tracked errors in the assigned values rather than averaging those errors over the entire range of the scale. By normalizing the primary standards on a linear system, using the full suite of primary standards multiple





times over several years (as was done for X2019), we can better define the assigned values of the primary standards. After converting to X2019, the NDIR system is still subject to end effects and errors in value assignments of the primary standards, but these errors are much smaller compared to X2007, and the comparison data show much better agreement between the two

systems (lower panel in Fig. 12). We expect future value assignments to be more consistent than those based on the NDIR method.

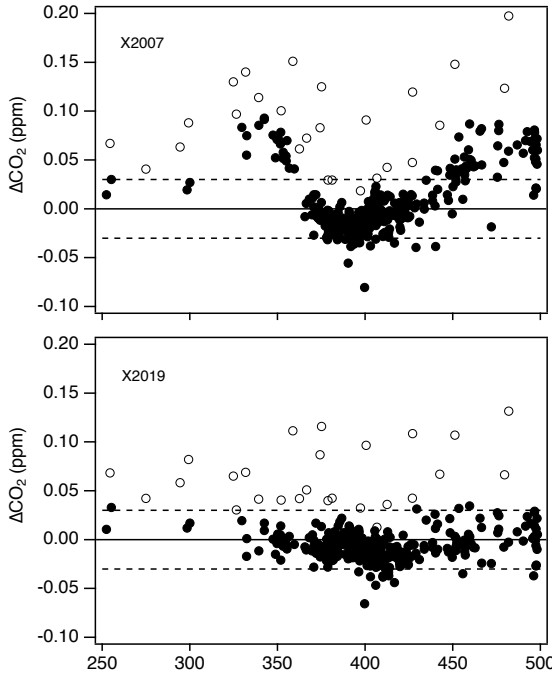

**Figure 12**: Differences between NDIR and laser spectroscopic systems used for tertiary value-assignment on X2007 (upper panel) and X2019 (lower panel) during a 12-month overlap period. Open symbols denote tertiary standards with significatly

lower $^{13}$C-$CO_2$ isotopic ratios compared to the others ($\delta^{13}$C < -20‰), and thus subject to bias in the NDIR measurement. Dashed lines are the expected reproducibility of the NDIR system (±0.03 ppm).

**9.1 Approximating X2019 using a linear scale conversion**

For users of standards obtained from the CCL, the best way to update to the X2019 scale is to implement the X2019 re-

assignments and propagate through to atmospheric data. A database management system allows for efficient propagation of scale changes to atmospheric data. However, for datasets in which a full reprocessing is not possible or practical, a linear scale conversion could be an option. The linear function shown below is based on primary standard assigned values (weighted linear regression):





$$X2019 = 1.00079 * X2007 - 0.142 \ (ppm) \qquad (6)$$

It is clear from Fig. 13 that the linear conversion, shown as the solid black line, will introduce errors in the 370-390 ppm range compared to full reprocessing, as the line does not pass through the majority of the data in that range. This is an unfortunate consequence of the mis-assigned primary standard (AL47-146) and the mis-assigned secondary in use in 2008. Nevertheless,

the linear conversion introduces errors less than 0.05 ppm for 94% of the tertiary standards in the range 320-460 ppm (Fig. 14). Errors are less than 0.03 ppm for 78% of the data in the same range, although there is a persistent low bias between 380 and 390 ppm.

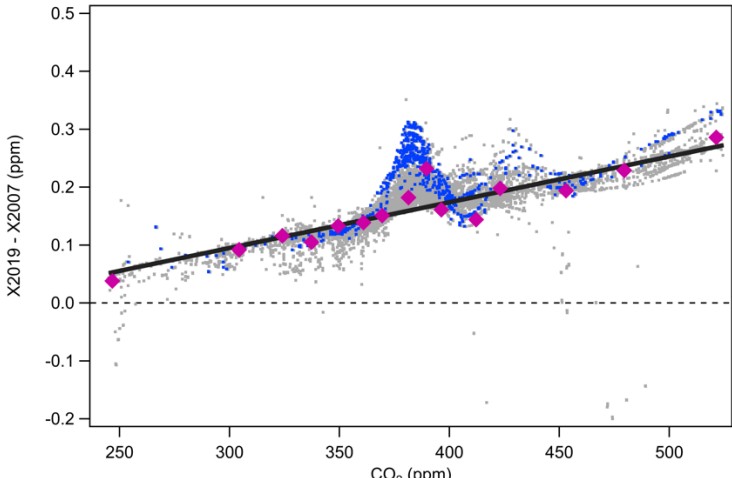

**Figure 13**: Differences between X2019 and X2007 tertiary assignments from 1995 to 2017, (NDIR only) showing 2008

analysis in blue, all others in gray. A linear scale conversion derived from primary standards (pink) is shown as the black line.

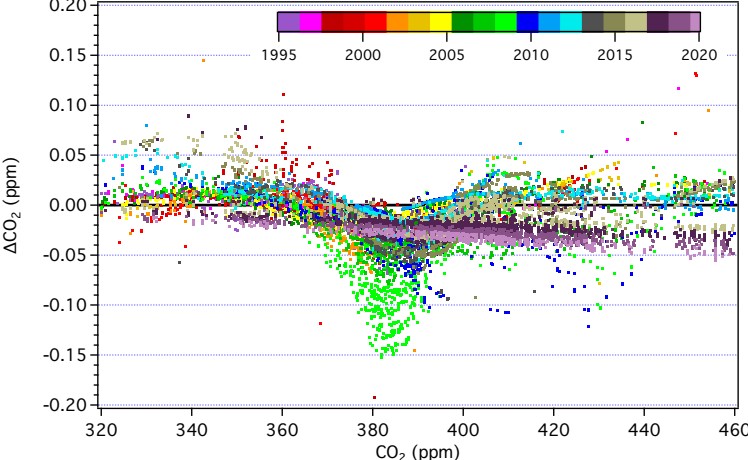

**Figure 14**: Scale conversion bias seen in tertiary standard assignments when using the linear scale conversion, shown as the difference between the linear scale conversion and the reprocessed values.



## 9.2 Revision of NOAA atmospheric data

We have reprocessed NOAA atmospheric data back to ~1979 for internal evaluation. This involved re-assigning $X_{CO2}$ values for working (tertiary level) standards to X2019 by reprocessing the original tertiary-secondary comparisons. For data prior to 1995, this also involved converting from a Scripps Institution of Oceanography (SIO) scale to X2019. Complete detail of the conversion from the SIO scale to X2019 is beyond the scope of this paper, and will be addressed in a separate publication. After fully converting to X2019, NOAA data prior to ~1979 will still be traceable to the SIO scale in use at the time of

measurement.

We include examples of atmospheric data here to provide a comparison of two methods used for propagating the X2019 scale: full reprocessing using updated tertiary standard values and response functions, and a simple linear scale conversion applied to atmospheric records. Actual bias introduced into atmospheric records by implementing the linear conversion will depend

on the calibration procedures used in a particular laboratory, and the range and calibration history of standards. For example, if a particular set of standards used by a laboratory was analyzed multiple times by the CCL over several years, the impact of the 2008-2009 secondary standard mis-assignment would be reduced.

The lower panel in Fig. 15 shows the difference between the linear scale conversion and full reprocessing applied to in situ

$CO_2$ at Mauna Loa, HI (MLO). Generally, the linear scale conversion is fairly close to the fully reprocessed data but has a negative bias which is larger during 2007-2009 due to the 2008 secondary mis-assignment issue. There are time periods of larger differences, such as in late 2014, due to a reassessment of drift in the working standards. In the case of the 2014 period, one of the working standards had a relatively large drift correction (0.2 ppm yr$^{-1}$, which is not common), but the drift correction was implemented on X2007 in a way that exaggerated the effect (this only applies to relatively few cylinders in 2014). Without

fully reprocessing, this error would be preserved in the data set.





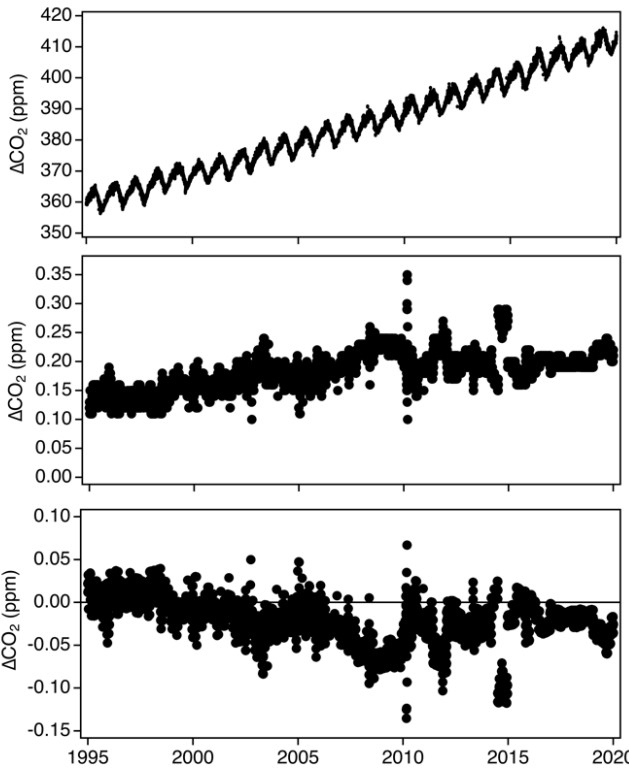

**Figure 15:** Hourly-averaged $CO_2$ measurements from Mauna Loa Observatory fully reprocessed to X2019 (upper panel), the difference between the fully reprocessed X2019 data and X2007 (middle panel), and the difference between using the linear scale conversion and full reprocessing methods to determine X2019 values (lower panel).


In addition to MLO, we reprocessed in situ data from the other NOAA baseline observatories (Barrow, AK; American Samoa; South Pole) and flask samples from marine boundary layer (MBL) sites using both the linear scale conversion and full reprocessing methods. Biases in the linear scale conversion were binned by year to get a sense of how well the linear scale conversion approximates the scale difference over time. Again, differences due to reassessment of drift in the working

standards are included in these binned bias terms. Fig. 16 shows the average annual bias in each of these data records that would be included if the records were converted to X2019 using the linear function rather than fully reprocessed (note, only hourly averages and flask samples identified as representing baseline conditions were used for this comparison). Average bias across the whole period is -0.03 ppm but there are years in individual records with biases up to -0.09 ppm. These measurement systems are tightly tied to the calibration chain. The larger biases during 2007–2009 show that these systems all follow the

bias in the scale due to the 2008-2009 mis-assigned secondary standard. The effect is moderated slightly due to the use of multiple standards and the fact that most standards have pre- and post-deployment value assignments and typically only one of these would have occurred during the 2008-2009 excursion.





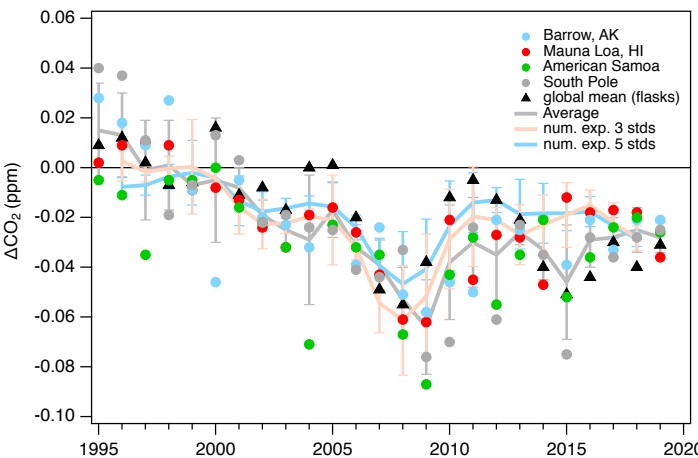

**Figure 16:** Estimates of scale conversion bias (linear scale conversion minus full reprocessing) derived from in situ
measurements at four NOAA observatories, global averages determined from measurements of discrete air samples collected
at marine boundary layer sites (gray line), and two numerical experiments (orange and blue lines, see text). All error bars are
one standard deviation. Numerical experimental results are shown as three-year running means.

We also conducted a numerical experiment to examine scale conversion bias without the added complications from a re-
assessment of drift in working standards. We randomly selected sets of three and five individual tertiary standards measured
within a calendar year. Each set required a standard within ±10 ppm of the global average from a particular year
(https://www.esrl.noaa.gov/gmd/ccgg/trends/global.html). The other standards were required to be at least 10 ppm but less
than 30 ppm apart and cover mole fractions above and below the initial selected standard. Quadratic fits to the actual X2019 -
X2007 differences vs. the X2007 assignments were made. The point on this curve corresponding to the calendar year global
average (on the X2007 scale) was compared to the global average converted to X2019 using the linear scale conversion. The
experiment was run 50 times for each year. In essence, this lets us approximate the bias due to the use of the linear scale
conversion on a hypothetical sample equal to the global average for 50 different sets of standards. The average biases due to
the use of the linear scale conversion for 3-standard and 5-standard suites are shown in Fig. 16 expressed as 3-year running
means. The results show good agreement with the bias seen in the in situ and flask MBL records. It is important to note that
both the results of the numerical experiment and these particular atmospheric records are tightly tied to the $CO_2$ scale transfer
system in time. Atmospheric data from 2007-2009 measured by external programs would not be as sensitive to the 2008 bias
if their standards were not calibrated by the CCL during that time. Conversely, measurements at other times tied to standards
that were only measured during the 2007-2008 period (without subsequent re-analysis) would be more sensitive.



### 9.3 Historical Scales

The impact of the revision from X2007 to X2019 is well understood and the linear conversion agrees with full reprocessing within 0.03 ppm for nearly 80% of standards value-assigned since 1995 over the range 320-460 ppm (Fig. 14). However, data traceable to NOAA scales prior to the release of X2007 that cannot be fully reprocessed are an additional concern. NOAA scales in use prior to X2007 were not identified by name, and involved a transition from scales based on SIO manometric measurements to NOAA manometric measurements. To assess the magnitude of potential bias relative to X2007 that could

exist in archived data sets still traceable to historical NOAA scales, we examined records from CSIRO (Australia), NIWA (New Zealand), and Environment Canada, who provided records of tertiary standard value-assignments prior to the formal adoption of the X2007 scale. Fig. 17 shows the difference between the original reported value (assigned by NOAA at that time) and the value re-assigned on scale X2007 upon its release.

NOAA primary standards were initially value-assigned by SIO from 1992 to 1995. From 1996-2000, we used a mixture of NOAA and SIO manometric results, and from 2001 onward we used only NOAA manometric results. Scales propagated by NOAA from 1993-2000 were effectively a mixture of the SIO scale in use at the time (now obsolete) and the NOAA manometric data up to that time. Bias is largest and shows more scatter prior to 1994 because the NOAA scale was based on relatively few SIO measurements of the NOAA primary standards (Guenther and Keeling, 2000). Primary assignments

improved over time as the number of measurements increased. Data traceable to these unnamed NOAA scales are biased relative to X2007 (Fig. 17). However, any potential bias in atmospheric records would be related to the date the standards were value-assigned, not necessarily the date the atmosphere was measured. The potential bias in historical data sets relative to X2019 would increase due to the X2019 to X2007 relationship. The linear conversion (equation 6) is not strictly applicable to data not traceable to X2007, but would be a close approximation for data traceable to scales in use between 2001 and 2006.

These limitations should be considered with regard to the uncertainty of historical data.





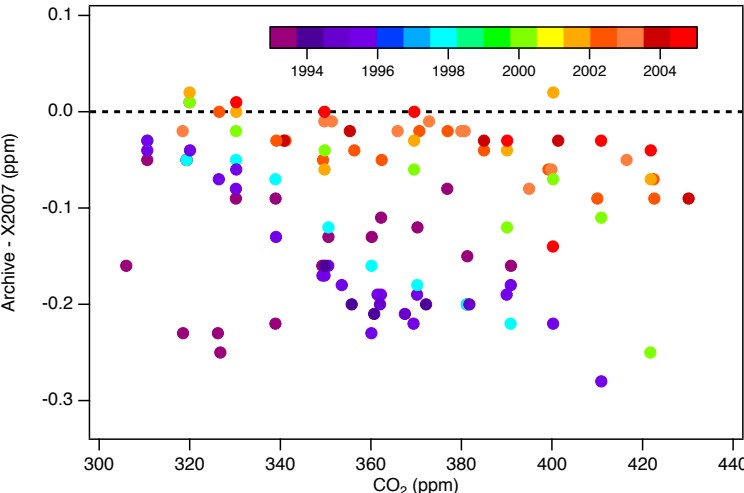

**Figure 17**: Potential bias that could exist in archive data sets traceable to NOAA standards prior to the release of X2007, shown as the difference between a hypothetical archived result and that result expressed on scale X2007 (derived from a sample of standards analyzed from 1993 to 2005).

## 10 Conclusions

We have applied two corrections to manometric data used to define the WMO/GAW $CO_2$ scale and include four additional standards to define a new scale, identified as WMO-$CO_2$-X2019. The net result of a scale update is two-fold: 1) The X2019 scale is more accurate and internally consistent than the previous X2007 scale. 2) Tertiary assignments on X2019 are more consistent across time because, with additional manometric analysis of primary standards and additional information on secondary assignments, scale propagation has been improved. While the scale difference at the tertiary standard level (~0.18 ppm at 400 ppm) is small in relative terms (0.045%), it is significant in terms of atmospheric monitoring. Measurement laboratories will need to update to the X2019 scale to avoid mis-interpretation of scale-induced (artificial), atmospheric gradients as real signals.

For users of standards obtained from the CCL, the best way to update to the X2019 scale is to implement the X2019 re-assignments and propagate through to atmospheric data. However, for datasets in which a full reprocessing is not possible or practical, a linear scale conversion is an option. The linear conversion will result in bias compared to full-reprocessing, but that bias is relatively small in many cases, and is less than 0.03 ppm for nearly 80% of standards value-assigned since 1995 over the range 320-460 ppm.



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

**Author contributions**

All authors played a role in experiment design and analysis. BH, AC, and PT performed the calculations. BH prepared the manuscript with contributions from all co-authors.

**Acknowledgments**

We appreciate the work of Conglong Zhao and Kirk Thoning, who developed the NOAA manometric method, and Ed Dlugokencky for thoughtful review of the manuscript. We thank Sylvia Michel and Bruce Vaughn at the University of Colorado Stable Isotope Laboratory for their analysis. We thank Paul Krummel and Ray Langenfelds of the Commonwealth Scientific and Industrial Research Organization (CSIRO) (Australia), Armin Jordan of the Max Planck Institute for Biogeochemistry (Jena, Germany), Doug Worthy (Environment Canada), and Gordon Brailsford at the National Institute for Water and Atmospheric Research (NIWA) (New Zealand) for providing historical calibration data for evaluation. This work was supported, in part, by the NOAA Atmospheric Chemistry, Carbon Cycle, and Climate Program. M. F. Schibig was supported by an Early Postdoc Mobility fellowship from the Swiss National Science Foundation.