# Peer review of "Revision of the WMO/GAW CO2 Calibration Scale"

_Atmospheric Measurement Techniques, 2020_

## Referee Comment (RC1) · Anonymous Referee #1 · 3 Dec 2020

The paper describes the basis for the revision the Central Calibration Laboratory is making to the WMO GAW CO2 calibration scale. It transparently explains a few issues that are impacting the currently employed WMO X2007 scale and that are the origin of a ca +0.2 ppm correction towards the new scale. An in-depth discussion of involved uncertainties and experimental limits is provided. Experiments and their results are described that were made to verify assumptions (on the influence of CO2 adsorption) and to validate the results (gravimmetric approach). Also results from comparison with NMIs are presented that provide reconfirming evidence for the assumed uncertainties. As the WMO scale has been the basis for all WMO GAW measurements the authors add a section that describes the implementation of the new scale to the measurement community. They offer a straightforward way to implement the scale for data sets where

a re-processing based on re-assignments of calibration gases is not assessible and provide information to estimate the associated bias.

General Comment

The topic is extremely relevant as it is the basis for all WMO GAW atmospheric measurements of the main greenhouse gas. It is obvious that the manuscript is based on rigorous work, analytic expertise and good bookkeeping. The quality of this work is exemplary. It is particularly commendable that the authors openly disclose lapses that had been made in the past and are rectified with the scale update and that they provide test results in the supplement material that might challenge their basic approach of a linear extrapolation of the CO2 loss rate to the start of the record. The paper is well written and definitely should be published in AMT.

There are some topics, though, where I would like to ask the authors for further clarification, completion or correction of the manuscript before publication.

Specific comments to the Manuscript text

l. 85: "During a measurement experiment, gas from a cylinder is loaded into the larger of the two volumes (large volume, ∼6 L). After flushing the large volume for 10 min. at 200 mL min-1 and allowing the gas temperature to equilibrate to oven temperature...". For me the reason for the 10 min flushing period that exchanges one third to one half of the large volume is not clear. I have come across the following statement made in Tans et al. 2017: "there may be a component due to gas handling issues on the NDIR system that cannot be resolved. This is still under investigation and will be addressed in a forthcoming paper discussing the scale revision" I have not been able to identify any section in this manuscript that has dealt with such issues. At the 0.01 ppm level pressure regulators are likely to introduce a bias in the CO2 mole fraction. There are certainly standardized gas handling procedures at the Central Calibration Laboratory. Yet, there are little details given in the manuscript.

l. 116 / l. 234: It is not explicitly stated if prior to each manommetric episode the volume ratio was redetermined using the gas expansion technique.

l. 132: Please add a reference to the Suppl. Material Figure S3.

l. 154 / Fig. 4: The text states that Fig. 4 shows the differences between original and updated results (implying that original > updated), the y-axis label is named "X_virial_correction". This seems as a contradiction in the algebraic sign. The text should state this more precisely.

l.160: In l. 235 it is written that the volume ratio changed again in 2014. Is it correct, that the correction function for beta for the period 2014-2016 is the same as 2004-2014?

l.199: It is not explained how the loss correction uncertainty is calculated. A reference to Suppl. Material section 2.4 (l.226) should be made.

l. 247 (Table 1): In the "CO2_primary_all_valid_data_X2019_supplement" file N = 34 x2019 values are listed for AL47-146 instead of N = 35 in Table 1 (one is flagged); for some cylinders the numbers in the Table 1 columns "Avg. (x2019)" and "s.d. (x2019)" are nearly but not exactly those that can be calculated based on the numbers provided in the "CO2_primary_all_valid_data_X2019_supplement" file.

l. 633: There had been a scale identified as X2005. (Tans, P. , Zhao, C. , and Thoning, K.: Revision of the International Calibration Scale for CO2-in-Air: WMO-X2005, 13th WMO/IAEA Meeting of Experts on Carbon Dioxide Concentration and Related Tracers Measurement Techniques (Boulder, Colorado, USA, 19-22 September 2005) WMO TD No. 1359, Geneva, Switzerland, 19-25, 2011). In this report the bias of old assignments (dating back to 1980) relative to the X2005 scale are presented with assignments from 1980-1995 being low by -0.2 ppm. At the 14th WMO/IAEA Meeting of Experts on Carbon Dioxide Concentration and Related Tracers Measurement Techniques in 2007 there has been a similar presentation by Pieter Tans showing X2007 – X2005 differences that indicated smaller mole fraction dependent differences ($\pm$ 0.05

ppm) that appeared stable over time (back to 1988)(see respective slides in Fig. 1-3). This is consistent to Fig. 17 but included many more data points.

There are few references in the text that do not appear in the reference list, namely:

l. 39 (WMO,2020); l. 126 (Sengers et al., 1971); l. 295 (White et al., 2015)

The references of Keeling et al 1986 and Tans et al. 2011 in the reference list are incomplete:

Keeling, C. D., Guenther, P. R., and Moss, D. J.: Scripps reference gas calibration system for carbon dioxide-in-air standards: revision of 1985, WMO;??, 1986

Tans, P. P., Zhao, C. L., and Kitzis, D.: The WMO Mole Fraction Scales for CO2 and other Greenhouse Gases, and Uncertainty of the Atmospheric Measurements, 15th WMO/IAEA Meeting of Experts on Carbon Dioxide, Other Greenhouse Gases and Related Tracers Measurement Techniques, GAW Report No. 194, Geneva, Switzerland, 152-159, 2011

Specific comments to the Supplemental Material text

1.1 Historical manometric records l. 15: "However, some manometer runs from 1998 and some from 2004 show two CO2 peaks (Fig. S1)". Please either omit CO2 or add "..show apparently two CO2 peaks.".

2.3 Volume Ratio

l. 123/equation (s3): The reader will not know Zhao et al. 1997 by heart. The equation (s3) is not self-explanatory as neither the definition of r1, r2, r3, r4 nor the derivation of the equation is given here. I suggest to shift the reference from l. 123 to below equation (s3) in a form like "(see Zhao et al. 1997 for further description of the volume ratios ri and the derivation of the (s3))."

l. 162-l.166: It is likewise not immediately obvious to the reader why it is relevant provide the uP for the specified pressures. It would help to add to l. 160: "for pressures

attained after expansion during volume ratio measurements r1, r2, r3, r4 as described in in Zhao et al. 1997" and add in each line e. g. l. 163: "On expansion to 19 kPa, (r1), 8 10-6 ..."

l. 171 / l. 190: It is not clear to me if the differences in volume ratio determinations that occur when expansion experiments are made with different gases are only due to uncertainties in the applied virial coefficient or if other experimental aspects are involved in using the three gases. In l. 171 the uncertainty estimate relates to n=3 (which appears to refer to the three data series presented in Fig. S6). The data basis for the larger uncertainty contribution quote 0.03 in l. 190 is not that clear.

l. 183: Table S2: Another column for the counter "i" = 1..4 would be useful.

l. 189: "We add to that uncertainty contributions from temperature probe placement (0.08)". Whereas 2.3.1 has covered this aspect it is not clear how eq. (s4) and (s5) translate to this uncertainty contribution term.

l. 192: In the main manuscript l. 269 only a subset of variables accounting for 0.012% of the uncertainty of the volume ratio is considered to be relevant for drift assessment. It would be useful to have this detailed in here.

2.4 Total uncertainty associated with the manometric measurement

Reference to equation (s8) is made in the lines 209, 219, 228, which should be corrected to (s6).

l. 223 & l. 226: Does the 15% uncertainty represent entirely the uncertainty of slope from the regression fit of the pressure data? The uncertainty of the loss correction depends on the uncertainty in the loss rate and the uncertainty in the time tmax_CO2 - t0. If the uncertainty of the loss rate itself is estimated to be 15% I would expect any further uncertainty in the duration would need to increase this percentage. A typical 0.015 ppm / 4cycles has been stated in the manuscript l. 187. It is not clear why the total loss correction uncertainty remains 15%.

l. 226: In l. 41-51 the potential influence of non-linearity in the CO2 sorption on the accuracy of a loss rate based on linear sorption behavior is considered. I think it is fair that the authors only consider an upper limit for this and conclude that the conditions under which they could make their experiment were not sufficiently representative for the manometric procedure as to derive any quantitative uncertainty contribution from it. However, I also think it would be fair to mention this here in the uncertainty section. that there is indication for a potential error in the loss rate determination. (something like "A potential bias resulting from a non-linear adsorption at the beginning of the experiment is assumed to be very small and could not be quantified experimentally")

Table S3: Please change Bair and BCO2 to betaair and betaCO2 (same as elsewhere in the manuscript); it is not clear from this manuscript how the beta standard uncertainty estimate of 0.2 is made. Please add the reference.

Table S3 is providing an overview of the various variables' uncertainty contributions and their effect on the total result of the manometric measurement. What I find incoherent and confusing, is to include the contents of the lowest three rows. As I understood the "Repeatability" is a standard error derived from the total set of manometric measurement results that have been made between 1996 and 2016 that define the X2019 scale. These measurement results are subject to the overall uncertainty involved in the manometric procedure (ïA¿0.079 ppm according to l. 229). In l. 238 it is explained that the episode results are deemed to be independent. Therefore, I would expect the scatter of the episode mean results to be in line with the estimated uncertainty (which is the case). What does not seem proper to me is to add the "Repeatability" term as an additional uncertainty contribution. It would appear more coherent if the Table S3 only contained the compilation of the manometry uncertainty terms (omitting the last three rows beginning with "13C,18O") and another table compiled the uncertainties associated with the measurement of a 400 ppm air sample. This latter then should include the standard error of the manometric measurements and the uncertainty of the scale transfer measurement including uncertainty associated with the stable isotope composition (note: this is discussed in the main paper but not mentioned in any part in the Suppl. Material document; a reference to section 6 of the paper would be helpful). The column "Approx. Relative contribution (%)" could be part of both tables, but should sum up to 100% in each of the tables.

2.5 Total uncertainty, including scale transfer

l. 237: I would assume "The manometer repeatability from up to 10 episodes" to be the average of column "Episode_std" in the "CO2_primary_all_valid_data_X2019_supplement" file, which is 0.077. Instead the authors take the s.d. (x2019) from Table 1, i.e. the standard deviation of the entire set of individual manometric measurements per cylinder. Therefore, please add for clarity: "The manometer repeatability from all individual manometric measurements made within up to 10 episodes is... 0.10 ppm (see Table 1 column "s.d.(x2019))". These represent measurements done over a period of 20 years with parts of the system having been replaced throughout this period and operators having changed. As episodes are also considered independent perhaps a change of the term "repeatability" to "reproducibility" would appropriate.

l. 239: The volume ratio value is also a critical component. I might have missed it but having read the paper it is not clear to me if the volume ratio value used for each episode is the result of an individual redetermination valid for the respective episode, or if the volume ratio numbers are the same for all episodes within the respective periods where the volume ratio was nominally the same 1996-1999, 1999-2004, 2004-2014, 2014-present. In the first case the uncertainty of the volume ratio determination would add to the scatter, in the latter case it would not (only changes of the manometry apparatus from episode to episode would). This should be stated somewhere in the manuscript.

l. 241: Which exact calculation results in 0.039 ppm? I could only get this result when dividing the values from Table 1 "s.d.(x2019)" by the square root of "Nep(x2019)"

the number of episodes, and averaging this for all cylinders listed in Table 1. This seems inconsistent to me: either the s.d. of the cylinders' episode means (16th column in the "CO2_primary_all_valid_data_X2019_supplement" file) should be divided by "Nep(x2019)" or else the "s.d.(x2019)" divided by the square root of "N(x2019)". Both would yield a smaller values, in the first case 0.028 ppm, in the latter 0.021 ppm.

l. 242ff: There has been changing analytical instrumentation over time as is described in section 9 of the manuscript. It should be repeated in l.242 that the uncertainty contribution from scale transfer with laser spectroscopic techniques does only apply to these and not to NDIR measurements. It would be good to have a statement here if the X2019 scale transfer uncertainty estimate for assignments that were made by NDIR before November 2016 remains as it has been estimated previously for the X2007 scale (0.034 ppm, k=2; https://www.esrl.noaa.gov/gmd/ccl/co2report.html). l. 247f: I cannot fully follow how Table S4 col. 3 is calculated. According to Table S3 for 400 ppm it should be $(0.079^2+0.039^2+0.01^2+0.01^2)0.5$ which is close to but not exactly 0.093 ppm.

Specific comments to the file "CO2_primary_all_valid_data_X2019_supplement"

Header: please add the formula how "Episode_unc" is computed. For me that is not obvious. It would be helpful if the "Cylinder#" would be harmonized with Table 1 "Cylinder" Cylinder 101 xdate has been filled out incorrectly for the June 2015 episode.

[Figure]

**Fig. 1.** Tans et al 2007: old assignments - X2005

[Figure]

[Figure]

**Fig. 2.** Tans et al 2007: X2005 - X2007 (times series)

[Figure]

**Fig. 3.** Tans et al 2007: X2005-X2007 (mole fraction dependence)

[Figure]

---

## Referee Comment (RC2) · Ray Langenfelds (Referee) · 7 Dec 2020

The Central Calibration Laboratory's role in maintaining and propagating the WMO CO2 scale is fundamental to the international atmospheric CO2 measurement effort. NOAA/GML have managed this responsibility over many years with skill and diligence. As a user of their services, I take this opportunity to thank them for their contribution.

This paper describes the latest revision of the scale. It builds on previous papers describing the calibration systems and earlier versions of the WMO scale. It gives a thorough account of the issues that necessitated this scale revision, evaluation of the growing body of historical data from the CCL's reference standards, comparison of X2007 and X2019 scales, uncertainty analysis etc. It is important that this informa-

tion be made available to the CO2 measurement community. Open acknowledgement of problems (occasional past mistakes, missing records, methodological limitations) adds to the value of the paper. This transparency is appreciated. The paper is very appropriate for AMT and can be published with only minor revisions.

Specific comments

1) Lines 69, 80, 81 – It would be more accurate to use the term "reference gases" instead of "reference materials", as the latter is commonly used for isotopically labelled materials including solids and liquids.

2) Line 86 – The brief description of the manometric procedure would be clearer by also stating that the larger volume is pre-evacuated, and its vacuum pressure.

3) Section 3.2 - The largest difference from the X2007 scale is allowance for loss of CO2 in the manometer o-rings. Direct tests of the manometric method suggest this process still leaves uncertainty of as much as 0.2 ppm in defining the absolute CO2 scale. It is sobering that uncertainties of this magnitude remain, when other metrics (e.g. network compatibility, scale propagation) can have much smaller uncertainty. The authors address this problem by comparing their manometric data with independent information, such as from gravimetrically derived scales (which have their own uncertainties), to arrive at a preferred definition of the manometric scale. I agree with their approach, though some questions remain over quantification of the uncertainty.

A key point that should be made in the paper is the distinction between total uncertainty in linking the scale to SI units and the component of uncertainty that pertains to maintaining a consistent scale and propagating it to other laboratories. The latter is more important for the CCL's main purpose of aligning CO2 data between laboratories. This concept has been recognised in earlier CCL papers but is not mentioned in this manuscript. Absolute accuracy of the scale is not critical for most applications, and can be revisited in future if the manometer o-ring effect becomes better quantified.

The brief description of the loss correction uncertainty in supplementary section 2.4 is not clear on whether it includes allowance for possible systematic error (elsewhere given an upper limit of 0.2 ppm) or random error only. This should be clarified.

A further minor point with the o-ring loss correction is the formulation in section 3.2. If "adsorption of CO2 begins about 1 minute after the liquid nitrogen is removed" and "there was a delay of about 2 minutes between the time the liquid nitrogen was removed and the first data record", why isn't the correction applied for an elapsed time of t_max_CO2 + 1 minute?

4) Section 6, p 16/17 - Primary standards were analysed on the laser-spectroscopy (LS) system 6 times over 3 years. Linear fits to these measurements against their manometric assignments yield residuals for individual standards that are highly consistent across the 6 episodes, and lie in a significantly large range of +/- 0.1 ppm with a standard deviation of 0.05 ppm (Figure 9). It is correctly stated that "variability seen in the residuals relates to the manometer average values". The residuals are then partly attributed to shorter manometric measurement histories for higher mixing ratio standards. Other sources of variance are implied but not specified.

An assumption of the calibration system (manometric + LS harmonization) is that these residuals represent random error in manometric average values. However, the standard deviation is about double that expected based on a 0.1 ppm (1-sigma) manometric uncertainty and the number of measurements listed in Table 1. This suggests some systematic bias between the techniques. Do the authors have any insight into possible causes?

My thoughts would be - firstly, can an isotopic bias be ruled out? Data in Table 2 suggest an isotopic bias is unlikely. Could there be sensitivity in either technique to other components of the gas matrix? If so, it is hard to see an unexplained sensitivity in the manometer given the documented evaluation of that technique. Maybe some gas handling bias? Depending on the cause(s) of the bias, has the uncertainty been

adequately captured?

5) There is no mention in the manuscript of gas handling effects or uncertainties, in particular regarding regulators. Are the author satisfied that the techniques referred to here are not subject to any significant gas handling effects, and/or can uncertainties be quantified? It was noted by Tans et al., 2017 that "Although there may be a component due to gas handling issues on the NDIR system that cannot be resolved. This is still under investigation and will be addressed in a forthcoming paper discussing the scale revision." Some comment on the current understanding of gas handling uncertainty is desirable.

Technical comments

75 – replace viral with virial

106 – fix inconsistent cold trap temperatures shown in the caption and figure legend

178 – replace were with where

377 – What is the basis of the NIST CO2 scale - gravimetric?

393, SM 193 – should read "Zhao and Tans (2006)"

394 – "JCGM, 2008" needs a reference or link

405 – should read "N2O is sufficient"

444 – "function of XCO2"

504 – "one analysis record"

521 – "due to the use"

681 – The Guenther and Keeling reference needs more information to be accessible. It does not appear to have further traceability details other than being a "technical report" but can be accessed online at https://scrippsco2.ucsd.edu/assets/publications/guenther_manometric_analysis_cdiac_report_2000.pdf

688 – Keeling, C.D., Guenther, P.R. and Moss, D.J., Scripps Reference Gas Calibration System for Carbon Dioxide-in-Air Standards: Revision of 1985. Environmental Pollution Monitoring and Research Programme No. 42, Technical Document WMO/TD-No. 125, 1986. https://scrippsco2.ucsd.edu/assets/publications/keeling_scripps_ref_gas_calibration_system_revision_1986.pdf

707 – Tans, P. P., Zhao, C. L., and Kitzis, D.: The WMO Mole Fraction Scales for CO2 and other greenhouse gases, and uncertainty of the atmospheric measurements, Report of the 15th WMO/IAEA Meeting of Experts on Carbon Dioxide, other Greenhouse Gases, and Related Measurement Techniques, 7–10 September 2009, GAW Report No. 194, WMO TD No. 1553, 152–159, 2011.

SM 51 – "as a way"

SM 209, 219, 228 – replace s8 with s6
* * *

---

## Author Comment (AC1) · 3 Feb 2021

doi:10.5194/amt-2020-408-RC1, 2020 Author Response to Reviewer 1 (RC1)

Thank you for a thorough review. We greatly appreciate the detailed comments and the time taken to check our calculations.

Specific comments to the Manuscript text l. 85: "During a measurement experiment, gas from a cylinder is loaded into the larger of the two volumes (large volume, âĹ̇ij6 L). After flushing the large volume for 10 min. at 200 mL min-1 and allowing the gas temperature to equilibrate to oven temperature...". For me the reason for the 10 min flushing period that exchanges one third to one half of the large volume is not clear. I have come across the following statement made in Tans et al. 2017: "there may be a

component due to gas handling issues on the NDIR system that cannot be resolved. This is still under investigation and will be addressed in a forthcoming paper discussing the scale revision" I have not been able to identify any section in this manuscript that has dealt with such issues. At the 0.01 ppm level pressure regulators are likely to introduce a bias in the CO2 mole fraction. There are certainly standardized gas handling procedures at the Central Calibration Laboratory. Yet, there are little details given in the manuscript.

Response: We have revised the text as follows.

During a measurement experiment, the manometer is evacuated to ∼5 mtorr and then gas from a cylinder is loaded into the larger of the two volumes (large volume, 6 L). The large volume is flushed for 10 min. at 200 mL min-1 and the exit gas stream is monitored by NDIR to ensure a stable CO2 signal. Inability to observe a stable CO2 signal (< 0.1 ppm) can result in the run being aborted. The large volume is then sealed off, allowed to equilibrate for five minutes, and the large volume temperature and pressure are recorded.

Other issues related to gas handling and regulators are addressed elsewhere. In the Supplement we include:

One aspect of the scale transfer not represented by TT (target tank) results is any impact that changing regulators would have since regulators are not typically removed from TT's to prevent damage to the cylinder valve fittings. For normal calibration services, a regulator is installed and conditioned following standard protocols (ref https://www.esrl.noaa.gov/gmd/ccl/reg.guide.html). Comparisons of pre- and post-deployment calibrations of standards used at NOAA sites, while complicated by drift issues during use, align with the expected reproducibility based on TT's. Regulators remain an issue requiring further investigations as the CCL attempts to improve calibration services.

l. 116 / l. 234: It is not explicitly stated if prior to each manometric episode the volume

ratio was redetermined using the gas expansion technique.

Response: This is now explicitly stated. Volume ratio experiments were performed prior to each episode and during an episode (e.g. Fig. S6 in Supplemental Material).

l. 132: Please add a reference to the Suppl. Material Figure S3.

Response: We have followed the suggestion.

l. 154 / Fig. 4: The text states that Fig. 4 shows the differences between original and updated results (implying that original > updated), the y-axis label is named "X_virial_correction". This seems as a contradiction in the algebraic sign. The text should state this more precisely.

Response: Thank you for pointing this out. We have fixed the text.

l.160: In l. 235 it is written that the volume ratio changed again in 2014. Is it correct, that the correction function for beta for the period 2014-2016 is the same as 2004-2014?

Response: Yes, a volume ratio change was made in 2014. However, we use the same correction function for over the period 2004-2016 because the volume ratio change was relatively minor and the correction function described the full periods fairly well.

l.199: It is not explained how the loss correction uncertainty is calculated. A reference to Suppl. Material section 2.4 (l.226) should be made.

Response: We now show in the header to the CO2_primary_all_valid_data_X2019_supplement how the loss correction uncertainty is calculated. And we clarified some text in the Supplement.

"For each measurement in the database, we calculate a loss correction and a virial correction. The uncertainty associated with Xvirial_correction is ~0.005 ppm. For Xloss_correction , we estimate the uncertainty in loss rate at 10% for most measurements, and 20% for those exhibiting a second maxima. We assume that the time corresponding to peak CO2 (t) is known to within one measurement cycle, and that the

initial time (to) has an uncertainty of 2 minutes (2-4 measurement cycles). Together, the uncertainty associated with the loss correction is ∼12% for most measurements, and 25-40% when a second CO2 maxima was observed. Although a potential bias resulting from a non-linear adsorption at the beginning of the experiment was observed in separate tests (fig. S3), the magnitude of this potential bias could not be quantified experimentally under conditions consistent with manometric experiments. "

l. 247 (Table 1): In the "CO2_primary_all_valid_data_X2019_supplement" file N = 34 x2019 values are listed for AL47-146 instead of N = 35 in Table 1 (one is flagged); for some cylinders the numbers in the Table 1 columns "Avg. (x2019)" and "s.d. (x2019)" are nearly but not exactly those that can be calculated based on the numbers provided in the "CO2_primary_all_valid_data_X2019_supplement" file.

Response: Thank you for catching the error for AL47-146 and checking data in the Supplemental data set. We did not check all averages and standard deviations listed in the supplemental data file at the time of submission. We did, however, check a number at random and did not find discrepancies. We should have mentioned in the header that for episodes with fewer than three runs on a given cylinder, we set the episode std dev to the actual std dev or the average std dev over all primary stds during that episode, whichever was larger. This avoids some instances with very small std dev on only two runs. Having now checked all entries, we have found a few discrepancies and made corrections. The updates were mainly to episode standard deviations and a few x2019 values. The errors in x2019 values apply only to the Supplement file and not to the values shown in Table 1. Changes to Episode_unc have a minor impact on the drift assessment. The conclusion remains the same, that we cannot detect drift within uncertainties.

l. 633: There had been a scale identified as X2005. (Tans, P. , Zhao, C. , and Thoning, K.: Revision of the International Calibration Scale for CO2-in-Air: WMO-X2005, 13th WMO/IAEA Meeting of Experts on Carbon Dioxide Concentration and Related Tracers Measurement Techniques (Boulder, Colorado, USA, 19-22 September 2005)

WMO TD No. 1359, Geneva, Switzerland, 19-25, 2011). In this report the bias of old assignments (dating back to 1980) relative to the X2005 scale are presented with assignments from 1980-1995 being low by -0.2 ppm. At the 14th WMO/IAEA Meeting of Experts on Carbon Dioxide Concentration and Related Tracers Measurement Techniques in 2007 there has been a similar presentation by Pieter Tans showing X2007 – X2005 differences that indicated smaller mole fraction dependent differences ($\pm$ 0.05 ppm) that appeared stable over time (back to 1988)(see respective slides in Fig. 1-3). This is consistent to Fig. 17 but included many more data points.

Response: While there have been discussions of named scales prior to X2007, such as X2005, the extent to which the X2005 scale was implemented retrospectively is not clear. It was customary in the 2000s to present manometric results at GGMT meetings. Following each episode, we averaged manometric data up to that point, performed harmonization experiments by NDIR, and compared recent results to previous results. However, implementation of the updated data (scales) was not rigorously documented prior to X2007. "

We have revised the text as follows:

The implementation of NOAA scales prior to X2007 was not rigorously documented. Prior to 2001, NOAA scales were partially based on SIO value assignments of the NOAA primary standards and thus were sensitive to revisions of the SIO scale. The incorporation of SIO revisions over time at NOAA and how these translated into distributed scales is not well documented, and therefore it is difficult to determine relationships between X2019 and historical scales prior to the full conversion to X2007. (Note that the CCL has taken multiple steps since then to ensure these lapses do not occur again and that the evolution of the scale is transparent and fully documented.)

There are few references in the text that do not appear in the reference list, namely: l. 39 (WMO,2020); l. 126 (Sengers et al., 1971); l. 295 (White et al., 2015)

Response: We have added the missing references. Thank you.

The references of Keeling et al 1986 and Tans et al. 2011 in the reference list are incomplete:

Response: We updated both references.

Keeling, C. D., Guenther, P. R., and Moss, D. J.: Scripps reference gas calibration system for carbon dioxide-in-air standards: revision of 1985, WMO/TD-125, 1986 Tans, P. P., Zhao, C. L., and Kitzis, D.: The WMO Mole Fraction Scales for CO2 and other Greenhouse Gases, and Uncertainty of the Atmospheric Measurements, 15th WMO/IAEA Meeting of Experts on Carbon Dioxide, Other Greenhouse Gases and Related Tracers Measurement Techniques, GAW Report No. 194, Geneva, Switzerland, 152-159, 2011 Specific comments to the Supplemental Material text

Historical manometric records l. 15: "However, some manometer runs from 1998 and some from 2004 show two CO2 peaks (Fig. S1)". Please either omit CO2 or add "..show apparently two CO2 peaks.".

Response: We have made the change. "However, some manometer runs from 1998 and 2004 show two peaks"

2.3 Volume Ratio

l. 123/equation (s3): The reader will not know Zhao et al. 1997 by heart. The equation (s3) is not self-explanatory as neither the definition of r1, r2, r3, r4 nor the derivation of the equation is given here. I suggest to shift the reference from l. 123 to below equation (s3) in a form like "(see Zhao et al. 1997 for further description of the volume ratios ri and the derivation of the (s3))."

Response: We have made the change as suggested. We also added a column to Table S2 identifying each volume expansion (1-4).

l. 162-l.166: It is likewise not immediately obvious to the reader why it is relevant provide the uP for the specified pressures. It would help to add to l. 160: "for pressures attained after expansion during volume ratio measurements r1, r2, r3, r4 as described

in in Zhao et al. 1997" and add in each line e. g. l. 163: "On expansion to 19 kPa, (r1), 8 10-6 ..."

Response: We have a change similar to your suggestion.

"Listed here are the components corresponding to uncertainties on each pressure measurement for the four successive volume expansions used to calculate the intermediate volume rations in equation (s3) (respectively: manufacturer's specification, zero drift, leak potential). Each volume expansion starts at 80 kPa."

l. 171 / l. 190: It is not clear to me if the differences in volume ratio determinations that occur when expansion experiments are made with different gases are only due to uncertainties in the applied virial coefficient or if other experimental aspects are involved in using the three gases. In l. 171 the uncertainty estimate relates to n=3 (which appears to refer to the three data series presented in Fig. S6). The data basis for the larger uncertainty contribution quote 0.03 in l. 190 is not that clear.

Response: You are correct, in that uncertainties in the second virial coefficients could partly explain the differences observed with different gases, but surface interactions may also play a role. We include the additional uncertainty derived from different gases because we see differences and do not fully understand them. We estimate this component from a uniform distribution over the full range of observations with three different gases (0.15/2/sqrt(3)) = 0.043.

l. 183: Table S2: Another column for the counter "i" = 1..4 would be useful. We added the column.

l. 189: "We add to that uncertainty contributions from temperature probe placement (0.08)". Whereas 2.3.1 has covered this aspect it is not clear how eq. (s4) and (s5) translate to this uncertainty contribution term.

Response: Thank you for catching that. This term was intended to capture the uncertainty related to temperature gradients in the oven and the variability observed using

different temperature probes. The value was overestimated and included elements already accounted for. On re-examination of the data, we increased slightly the uncertainties used in the volume expansion calculation (from 0.015 to 0.02 deg), and include only a small additional term (0.01) to account for small variations in volume that would results from calculating the VR using different combinations of temperature probes. Note that this results in a lower overall uncertainty for the volume ratio and a slightly lower overall uncertainty on the $CO_2$ mole fraction estimate.

The text now reads:

"The uncertainty calculated from parameters in Table S2 is 0.115 for a volume ratio of 880. Adding repeatability (0.032) in quadrature we obtain 0.119. Additional terms associated with PRT placement (0.01) and different gases (0.043) are included in a further step. The oven contains three PRTs but only two are used for the calculation of VR, and we get slightly different results using different combinations of PRTs. We also calculate different volume ratios using different gases (air, nitrogen, and argon). These differences could be partly related to uncertainty in virial coefficients, but could also involve surface interactions. For this component we assume a uniform distribution over the range 0.15 (0.15/2/sqrt(3) = 0.043). Summing all terms in quadrature we obtain: $u\Phi$ = sqrt(0.1152+0.0322+0.012+0.0432) = 0.127 for a volume ratio of 880.1, or 0.014%. This uncertainty estimate is about 40% larger than that reported by Zhao and Tans (2006). Since the last two uncertainty components are meant to capture elements that would be common to all volume ratio experiments, these are not included in the uncertainty applied to the drift assessment. "

l. 192: In the main manuscript l. 269 only a subset of variables accounting for 0.012% of the uncertainty of the volume ratio is considered to be relevant for drift assessment. It would be useful to have this detailed in here.

Response: We have attempted to clarify. On revision of the volume ratio uncertainty, this separate accounting does not impact the conclusion of the drift assessment (see

above).

2.4 Total uncertainty associated with the manometric measurement: Reference to equation (s8) is made in the lines 209, 219, 228, which should be corrected to (s6).

We have made the correction.

l. 223 & l. 226: Does the 15% uncertainty represent entirely the uncertainty of slope from the regression fit of the pressure data? The uncertainty of the loss correction depends on the uncertainty in the loss rate and the uncertainty in the time tmax_CO2 - t0. If the uncertainty of the loss rate itself is estimated to be 15% I would expect any further uncertainty in the duration would need to increase this percentage. A typical 0.015 ppm / 4cycles has been stated in the manuscript l. 187. It is not clear why the total loss correction uncertainty remains 15%.

Response: The original text did not describe our estimate correctly. We do, indeed account for the uncertainty in tmax_CO2 and t0. We have revised the text as follows.

"For Xloss_correction , we estimate the uncertainty in loss rate at 10% for most measurements, and 20% for those exhibiting a second maxima. We assume that the time corresponding to peak CO2 (t) is known to within one measurement cycle, and that the initial time (to) has an uncertainty of 2 minutes (2-4 measurement cycles). Together, the uncertainty associated with the loss correction is $\sim$12% for most measurements, and 25-40% when a second CO2 maxima was observed. Although a potential bias resulting from a non-linear adsorption at the beginning of the experiment was observed in separate tests (fig. S3), the magnitude of this potential bias could not be quantified experimentally under conditions consistent with manometric experiments. "

l. 226: In l. 41-51 the potential influence of non-linearity in the CO2 sorption on the accuracy of a loss rate based on linear sorption behavior is considered. I think it is fair that the authors only consider an upper limit for this and conclude that the conditions under which they could make their experiment were not sufficiently representative for

the manometric procedure as to derive any quantitative uncertainty contribution from it. However, I also think it would be fair to mention this here in the uncertainty section. that there is indication for a potential error in the loss rate determination. (something like "A potential bias resulting from a non-linear adsorption at the beginning of the experiment is assumed to be very small and could not be quantified experimentally")

Response: Thank you for this suggestion. See revised text above.

Table S3: Please change Bair and BCO2 to betaair and betaCO2 (same as elsewhere in the manuscript); it is not clear from this manuscript how the beta standard uncertainty estimate of 0.2 is made. Please add the reference.

Response: We made the suggested changes.

Table S3 is providing an overview of the various variables' uncertainty contributions and their effect on the total result of the manometric measurement. What I find incoherent and confusing, is to include the contents of the lowest three rows. As I understood the "Repeatability" is a standard error derived from the total set of manometric measurement results that have been made between 1996 and 2016 that define the X2019 scale. These measurement results are subject to the overall uncertainty involved in the manometric procedure (iÌĹA ÌÍ£0.079 ppm according to l. 229). In l. 238 it is explained that the episode results are deemed to be independent. Therefore, I would expect the scatter of the episode mean results to be in line with the estimated uncertainty (which is the case). What does not seem proper to me is to add the "Repeatability" term as an additional uncertainty contribution. It would appear more coherent if the Table S3 only contained the compilation of the manometry uncertainty terms (omitting the last three rows beginning with "13C,18O") and another table compiled the uncertainties associated with the measurement of a 400 ppm air sample. This latter then should include the standard error of the manometric measurements and the uncertainty of the scale transfer measurement including uncertainty associated with the stable isotope composition (note: this is discussed in the main paper but not mentioned in any part in the

Suppl. Material document; a reference to section 6 of the paper would be helpful). The column "Approx. Relative contribution (%)" could be part of both tables, but should sum up to 100% in each of the tables.

Response: We agree that the Table as presents was confusing. We have revised the uncertainty table into two sections: one showing contributions related only to the manometric measurement (with the total). We then add to this reproducibility and scale transfer uncertainty. We hope this is more clear. Since the scale transfer components (reproducibility of the laser system and uncertainty associated with isotopic differences) are small, we do not include those in the relative uncertainty estimates. 2.5 Total uncertainty, including scale transfer

l.     237:     I    would     assume    "The    manometer    repeatability    from    up to 10 episodes" to be the average of column "Episode_std" in the "CO2_primary_all_valid_data_X2019_supplement" file, which is 0.077.     Instead the authors take the s.d. (x2019) from Table 1, i.e. the standard deviation of the entire set of individual manometric measurements per cylinder. Therefore, please add for clarity: "The manometer repeatability from all individual manometric mea- surements made within up to 10 episodes is...  0.10 ppm (see Table 1 column "s.d.(x2019))". These represent measurements done over a period of 20 years with parts of the system having been replaced throughout this period and operators having changed. As episodes are also considered independent perhaps a change of the term "repeatability" to "reproducibility" would appropriate.

Response: See above. We have made changes to the uncertainty discussion, and use the term reproducibility, as suggested.

l. 239: The volume ratio value is also a critical component. I might have missed it but having read the paper it is not clear to me if the volume ratio value used for each episode is the result of an individual redetermination valid for the respective episode, or if the volume ratio numbers are the same for all episodes within the respective periods

where the volume ratio was nominally the same 1996-1999, 1999-2004, 2004-2014, 2014-present. In the first case the uncertainty of the volume ratio determination would add to the scatter, in the latter case it would not (only changes of the manometry apparatus from episode to episode would). This should be stated somewhere in the manuscript.

Response: We now explicitly state that the volume ratio is calculated before and during each episode.

l. 241: Which exact calculation results in 0.039 ppm? I could only get this result when dividing the values from Table 1 "s.d.(x2019)" by the square root of "Nep(x2019)" the number of episodes, and averaging this for all cylinders listed in Table 1. This seems inconsistent to me: either the s.d. of the cylinders' episode means (16th column in the "CO2_primary_all_valid_data_X2019_supplement" file) should be divided by "Nep(x2019)" or else the "s.d.(x2019)" divided by the square root of "N(x2019)". Both would yield a smaller values, in the first case 0.028 ppm, in the latter 0.021 ppm.

Response: See above. The value 0.039 was incorrect. The average standard error among episodes is 0.044 ppm.

l. 242ff: There has been changing analytical instrumentation over time as is described in section 9 of the manuscript. It should be repeated in l.242 that the uncertainty contribution from scale transfer with laser spectroscopic techniques does only apply to these and not to NDIR measurements. It would be good to have a statement here if the X2019 scale transfer uncertainty estimate for assignments that were made by NDIR before November 2016 remains as it has been estimated previously for the X2007 scale (0.034 ppm, k=2; https://www.esrl.noaa.gov/gmd/ccl/co2report.html).

Response: We have also expanded our discussion of scale transfer uncertainty, since this is particularly relevant for WMO-GAW. We will also be updating the CCL website. The page www.esrl.noaa.gov/gmd/ccl/co2report.html is out of date. Additional analysis suggests that the NDIR scale transfer uncertainty is 0.03 (1-sigma).

l. 247f: I cannot fully follow how Table S4 col. 3 is calculated. According to Table S3 for 400 ppm it should be (0.0792+0.0392+0.012+0.012)0.5 which is close to but not exactly 0.093 ppm.

Response: We have revised this section. We agree that the numbers did not quite add up.

Specific comments to the file "CO2_primary_all_valid_data_X2019_supplement"

Header: please add the formula how "Episode_unc" is computed. For me that is not obvious. It would be helpful if the "Cylinder#" would be harmonized with Table 1 "Cylinder" Cylinder 101 xdate has been filled out incorrectly for the June 2015 episode.

Response: We have made the suggested changes. The formula for Episode_unc is now included in the header. Thank you for finding typos.

Additional changes to manuscript:

We updated Table 2, now showing values to 3 decimal places, since this is how values are used for scale propagation.

---

## Author Comment (AC2) · 3 Feb 2021

doi:10.5194/amt-2020-408-RC1, 2020 Author Response to Reviewer 2 (RC2)

Thank you for a thorough review. We greatly appreciate the detailed comments and the time taken to check our calculations.

Specific comments 1) Lines 69, 80, 81 – It would be more accurate to use the term "reference gases" instead of "reference materials", as the latter is commonly used for isotopically labelled materials including solids and liquids.

Response: While we would contend that reference material is an appropriate term, we have changed the wording to reference gases, as this term is probably more familiar to the WMO/GAW community.

[Figure]

2) Line 86 – The brief description of the manometric procedure would be clearer by also stating that the larger volume is pre-evacuated, and its vacuum pressure.

Response: We have made the suggested changes.

The text now reads: During a measurement experiment, the manometer is evacuated to $\sim$ 5 mtorr and then gas from a cylinder is loaded into the larger of the two volumes (large volume, $\sim$6 L). The large volume is flushed for 10 min. at 200 mL min-1 and the exit gas stream is monitored by NDIR to ensure a stable $CO_2$ signal. The large volume is then sealed off, allowed to equilibrate for five minutes, and the large volume temperature and pressure are recorded.

3) Section 3.2 - The largest difference from the X2007 scale is allowance for loss of $CO_2$ in the manometer o-rings. Direct tests of the manometric method suggest this process still leaves uncertainty of as much as 0.2 ppm in defining the absolute $CO_2$ scale. It is sobering that uncertainties of this magnitude remain, when other metrics (e.g. network compatibility, scale propagation) can have much smaller uncertainty. The authors address this problem by comparing their manometric data with indepen- dent informa- tion, such as from gravimetrically derived scales (which have their own uncertainties), to arrive at a preferred definition of the manometric scale. I agree with their approach, though some questions remain over quantification of the uncertainty. A key point that should be made in the paper is the distinction between total uncer- tainty in linking the scale to SI units and the component of uncertainty that pertains to maintaining a consistent scale and propagating it to other laboratories. The latter is more important for the CCL's main purpose of aligning $CO_2$ data between laboratories. This concept has been recognised in earlier CCL papers but is not mentioned in this manuscript. Absolute accuracy of the scale is not critical for most applications, and can be revisited in future if the manometer o-ring effect becomes better quantified. Re- sponse: We agree that more emphasis should be placed on scale transfer uncertainty. We have added several paragraphs and two tables to the Supplement to document our evidence for scale transfer uncertainty, both with respect to the laser spectroscopic

system and the previous NDIR system. The brief description of the loss correction uncertainty in supplementary section 2.4 is not clear on whether it includes allowance for possible systematic error (elsewhere given an upper limit of 0.2 ppm) or random error only. This should be clarified.

Response: We have revised the discussion on loss correction uncertainty.

It now reads: For Xloss_correction , we estimate the uncertainty in loss rate at 10% for most measurements, and 20% for those exhibiting a second maxima. We assume that the time corresponding to peak $CO_2$ (t) is known to within one measurement cycle, and that the initial time (to) has an uncertainty of 2 minutes (2-4 measurement cycles). Together, the uncertainty associated with the loss correction is ∼12% for most measurements, and 25-40% when a second $CO_2$ maxima was observed. Although a potential bias resulting from a non-linear adsorption at the beginning of the experiment was observed in separate tests (fig. S3), the magnitude of this potential bias could not be quantified experimentally under conditions consistent with manometric experiments.

A further minor point with the o-ring loss correction is the formulation in section 3.2. If "adsorption of $CO_2$ begins about 1 minute after the liquid nitrogen is removed" and "there was a delay of about 2 minutes between the time the liquid nitrogen was removed and the first data record", why isn't the correction applied for an elapsed time of t_max_CO2 + 1 minute?

Response: Apologies. We stated this incorrectly. There is a 3-minute (180 sec) delay in the software, which why we selected 2 minutes for the delay in reprocessing. The text has been revised as: Adsorption of $CO_2$ probably begins about 1 minute after the liquid nitrogen is removed. For many data records, we know that there was a software delay of three minutes between the time the small volume was sealed off (and the liquid N2 removed) and the first data record. While this cannot be confirmed for all records, we include a two-minute delay: t_max_CO2 + 2 minutes (to = 2 min.).

4) Section 6, p 16/17 - Primary standards were analyzed on the laser-spectroscopy

(LS) system 6 times over 3 years. Linear fits to these measurements against their manometric assignments yield residuals for individual standards that are highly consistent across the 6 episodes, and lie in a significantly large range of +/- 0.1 ppm with a standard deviation of 0.05 ppm (Figure 9). It is correctly stated that "variability seen in the residuals relates to the manometer average values". The residuals are then partly attributed to shorter manometric measurement histories for higher mixing ratio standards. Other sources of variance are implied but not specified. An assumption of the calibration system (manometric + LS harmonization) is that these residuals represent random error in manometric average values. However, the standard deviation is about double that expected based on a 0.1 ppm (1-sigma) manometric uncertainty and the number of measurements listed in Table 1. This suggests some systematic bias between the techniques. Do the authors have any insight into possible causes? My thoughts would be - firstly, can an isotopic bias be ruled out? Data in Table 2 suggest an isotopic bias is unlikely. Could there be sensitivity in either technique to other components of the gas matrix? If so, it is hard to see an unexplained sensitivity in the manometer given the documented evaluation of that technique. Maybe some gas handling bias? Depending on the cause(s) of the bias, has the uncertainty been adequately captured?

Response: While some residuals seem larger than one might expect after averaging over 20 years, the standard deviations of the manometric histories are ~0.1 ppm. And because the volume ratio is partially common to all measurements, one cannot assume complete independence. Most residuals are 0.05 ppm or less, which is encouraging. However, the 521 ppm cylinder is something of an outlier with a residual of 0.1 ppm. We also wonder if gas handling may have some impact, e.g. the manometric analysis uses considerably more gas than NDIR or laser-spectroscopy. A typical manometric experiment consumes ~10L of gas (not counting flushing of regulators beforehand). Could this influence how $CO_2$ interacts with surfaces in the regulator, or the valve and neck of the cylinder (e.g. in the vicinity of the thread lubricant)? Perhaps. We do not have sufficient information to address these issues, but perhaps with modern high

precision analytical methods these issues can be explored more thoroughly. We do not think isotopic differences are a significant factor, but it is also true that the 521 ppm cylinder is the most isotopically different, and contains the largest "spike" of $CO_2$ injected on filling (in the 1990s). Nevertheless, the value of defining a scale based on a large number of standards is that some of the effects should get averaged out.

5) There is no mention in the manuscript of gas handling effects or uncertainties, in particular regarding regulators. Are the author satisfied that the techniques referred to here are not subject to any significant gas handling effects, and/or can uncertainties be quantified? It was noted by Tans et al., 2017 that "Although there may be a component due to gas handling issues on the NDIR system that cannot be resolved. This is still under investigation and will be addressed in a forthcoming paper discussing the scale revision." Some comment on the current understanding of gas handling uncertainty is desirable.

Response: We have added some text on this topic, although we do not have sufficient information to draw definitive conclusions. We added the following, along with supporting data in two tables.

"One aspect of the scale transfer not represented by TT (target tank) results is any impact that changing regulators would have since regulators are not typically removed from TT's to prevent damage to the cylinder valve fittings. For normal calibration services, a regulator is installed and conditioned following standard protocols (ref https://www.esrl.noaa.gov/gmd/ccl/reg.guide.html). Comparisons of pre- and post-deployment value assignments of standards used at NOAA sites, while complicated by drift issues during use, align with the expected reproducibility based on TT's. Regulators remain an issue requiring further investigations as the CCL attempts to improve calibration services."

Technical comments 75 – replace viral with virial - fixed 106 – fix inconsistent cold trap temperatures shown in the caption and figure legend – fixed 178 – replace were with where - fixed 377 – What is the basis of the NIST CO2 scale - gravimetric?, yes it is gravimetric, we added this to the text 393, SM 193 – should read "Zhao and Tans (2006)" - fixed 394 – "JCGM, 2008" needs a reference or link – we added a proper reference 405 – should read "N2O is sufficient" - - fixed 444 – "function of XCO2" - fixed 504 – "one analysis record" - fixed 521 – "due to the use" - fixed 681 – The Guenther and Keeling reference needs more information to be accessible.  It does not appear to have further traceability de- tails other than being a "technical report" but can be accessed online at https://scrippsco2.ucsd.edu/assets/publications/guenther_manometric_analysis_cdiac_report_2000.pdf

Response: We replaced this citation with Keeling et al. 2012, which is accessible on the SIO website. Keeling, R. F., Guenther, P. R., Walker, S., and Moss, D.: Scripps Reference Gas Calibration System for Carbon Dioxide-in-Nitrogen and Carbon Dioxide-in-Air Standards: Revision of 2012, Technical Report, March 2016, Scripps Insitution of Oceanography, La Jolla, CA, 2012.

688 – Keeling, C.D., Guenther, P.R. and Moss, D.J., Scripps Refer- ence Gas Calibration System for Carbon Dioxide-in-Air Standards:  Re- vision of 1985.  Environmental Pollution Monitoring and Research Pro- gramme No.  42, Technical Document WMO/TD-No.  125, 1986. https://scrippsco2.ucsd.edu/assets/publications/keeling_scripps_ref_gas_calibration_system_revision_1986.pdf

Response: We made the change to WMO/TD-No. 125 and added the URL

707 – Tans, P. P., Zhao, C. L., and Kitzis, D.: The WMO Mole Fraction Scales for CO2 and other greenhouse gases, and uncertainty of the atmospheric measurements, Report of the 15th WMO/IAEA Meeting of Experts on Carbon Dioxide, other Greenhouse Gases, and Related Measurement Techniques, 7–10 September 2009, GAW Report No. 194, WMO TD No. 1553, 152–159, 2011.

Response: Updated as suggested

SM 51 – "as a way" - fixed

SM 209, 219, 228 – replace s8 with s6 – fixed

Additional changes to manuscript: We updated Table 2, now showing values to 3 decimal places, since this is how values are used for scale propagation.